# West African–South American pandemic *Vibrio cholerae* encodes multiple distinct phage defence systems

David W. Adams [1,2] ✉, Milena Jaskólska [1], Alexandre Lemopoulos [1], Sandrine Stutzmann[1], Laurie Righi[1], Loriane Bader[1] & Melanie Blokesch [1,2] ✉

Our understanding of the factors underlying the evolutionary success of different lineages of pandemic *Vibrio cholerae* remains incomplete. The West African–South American (WASA) lineage of *V. cholerae*, responsible for the 1991–2001 Latin American cholera epidemic, is defined by two unique genetic signatures. Here we show that these signatures encode multiple distinct anti-phage defence systems. Firstly, the WASA-1 prophage encodes an abortive-infection system, WonAB, that renders the lineage resistant to the major predatory vibriophage ICP1, which, alongside other phages, is thought to restrict cholera epidemics. Secondly, a unique set of genes on the *Vibrio* seventh pandemic island II encodes an unusual modification-dependent restriction system targeting phages with modified genomes, and a previously undescribed member of the Shedu defence family that defends against vibriophage X29. We propose that these anti-phage defence systems likely contributed to the success of a major epidemic lineage of the ongoing seventh cholera pandemic.

The ability of lytic bacteriophages to rapidly devastate bacterial populations has driven the evolution of multiple layers of defence, including a diverse array of specialized anti-phage defence systems[1]. Bacteriophages also have the potential to affect bacterial pathogenesis, as exemplified by the cholera-toxin-encoding prophage CTXΦ (ref. 2). Moreover, bacteriophage predation is thought to limit the duration and severity of cholera epidemics and to affect individual patient outcomes[3,4]. Sampling of cholera patients has revealed that the O1 El Tor strains of *Vibrio cholerae* responsible for the ongoing seventh cholera pandemic (7PET) consistently co-occur with three lytic phages, ICP1, ICP2 and ICP3, with ICP1 being most frequently isolated[5,6]. Furthermore, the demonstration that a cocktail of ICP1, ICP2 and ICP3 can prevent cholera in animal infection models has led to renewed interest in using phages as a prophylactic treatment[7]. There is therefore an urgent need to understand the mechanisms by which *V. cholerae* defends against these and other viruses.

Since 1961, 7PET strains have spread out from the Bay of Bengal in a series of three distinct but overlapping waves[8]. Importantly, from wave 2 onwards, strains acquired SXT/R391 integrative and conjugative elements (SXT-ICE), which, in addition to carrying multiple antibiotic resistance genes, encode a variable anti-phage defence hotspot that in the globally dominant SXT-ICE harbours either BREX or restriction–modification systems active against ICP1–3 (refs. 8,9). In addition, a family of viral satellites, the phage-inducible chromosomal island-like elements (PLE), also occur sporadically and specifically parasitize ICP1 infection to mediate their own transmission[10,11]. Notably, these elements exemplify the co-evolutionary arms race of defence and counter-defence between bacterial hosts and their viral predators, with the emergence of resistant ICP1 phages that can overcome these defence mechanisms selecting for either alternative SXT-ICE carrying new defence systems or new PLE variants[9,12]. By contrast, how 7PET strains that lack SXT-ICE and PLE defend against these phages remains unknown. Given their ubiquitous nature and ability to shape epidemics, we hypothesized that these strains likely contained additional defence systems with activity against ICP1–3.

[1]Laboratory of Molecular Microbiology, Global Health Institute, School of Life Sciences, Ecole Polytechnique Fédérale de Lausanne (EPFL), Lausanne, Switzerland. [2]These authors contributed equally: David W. Adams, Melanie Blokesch. ✉e-mail: david.adams@epfl.ch; melanie.blokesch@epfl.ch

## Results

### WASA-1 prophage renders Peruvian strains resistant to ICP1

To test this hypothesis, we challenged a set of intensively studied wave 1 strains that lack SXT-ICE and PLE with a well-characterized collection of ICP1–3 phages that use the O1-antigen (ICP1 and ICP3) and OmpU (ICP2) as receptors[5,13,14]. All strains showed the expected sensitivities to ICP2 and ICP3 (Fig. 1a and Extended Data Fig. 1a). Unexpectedly, however, the widely used model strain A1552—a Peruvian isolate—was completely resistant to all 16 tested ICP1 isolates spanning a 27-year period, including the most recent isolates from the Democratic Republic of the Congo[15], showing a $10^5$-fold reduction in plaque formation (Fig. 1a and Extended Data Fig. 1a–e). Notably, the contemporary Peruvian strains C6706 and C6709 also behaved similarly. Interestingly, although zones of lysis were still observed at the highest concentrations of phage tested, re-streak tests revealed that these spots contained little to no viable particles (Extended Data Fig. 1c), suggesting that ICP1 propagation is substantially impaired and that these lysis zones represent 'lysis from without'[16]. Indeed, liquid replication assays comparing A1552 with the non-Peruvian strain E7946 showed that although genome injection proceeded similarly in both strains, in contrast to the robust replication observed with E7946, ICP1 was completely unable to replicate on strain A1552 (Extended Data Fig. 1d). ICP1 resistance in strain A1552 was independent of the *Vibrio* pathogenicity islands (VPI-1 and VPI-2) and the *Vibrio* seventh pandemic islands (VSP-I and VSP-II), which, together, harbour multiple known antiviral defence systems[17–23] (Extended Data Fig. 1e). Moreover, the ICP1-resistant strains encode the same repertoire of known defence systems as the ICP1-sensitive strains (Supplementary Table 1), suggesting that an as yet unrecognized defence system(s) is likely responsible for ICP1 resistance.

The Peruvian strains are members of the West African–South American (WASA) lineage of 7PET *V. cholerae*[8], which was responsible for a massive cholera epidemic that started in Peru in 1991, before spreading rapidly throughout Latin America, resulting in over 1.2 million cholera cases and approximately 12,000 deaths[24]. Interestingly, phylogenetic analysis has shown that these strains originated in West Africa, where they acquired two genetic signatures that define the WASA lineage: (1) a different set of genes on VSP-II and (2) the WASA-1 prophage[8,25–28]. As the deletion of VSP-II had no effect on ICP1 resistance, we tested strains deleted for WASA-1. Strikingly, this abolished defence against ICP1 in all tested Peruvian strains and restored ICP1 replication to levels similar to those of a susceptible control (Fig. 1b–e and Extended Data Fig. 1b). Moreover, introducing WASA-1 into non-WASA lineage strains rendered them ICP1 resistant (Extended Data Fig. 1f).

### WonAB system is responsible for ICP1 defence

The WASA-1 prophage is highly conserved throughout all WASA lineage strains[8,27] and, interestingly, is also found in diverse *Vibrio* species (Fig. 1b, Extended Data Fig. 2a and Supplementary Table 2). WASA-1 shows a conserved core genome encoding proteins involved in capsid and tail biogenesis and DNA replication, and a recombinase (Fig. 1b). To determine the gene(s) responsible for ICP1 defence, we used an existing RNA-sequencing dataset[29] to identify WASA-1 genes expressed under laboratory conditions and then screened truncations of these regions for loss of protection (Extended Data Fig. 2b). This revealed a two-gene operon (A1552VC_01233-34) encoding an ATPase and a hypothetical protein, downstream of a conserved putative transcriptional regulator (A1552VC_01232) (Fig. 1b). For the reasons outlined below, we have renamed this operon WASA overcoming lysogenization defect (OLD)–ATP-binding cassette (ABC) ATPase nuclease (WonAB). Strikingly, deletion of *wonAB* abolished protection against ICP1. Moreover, ectopic expression of *wonAB* complemented the Δ*wonAB* deletion and conferred complete protection to ICP1-sensitive non-WASA strains (Fig. 1d,e). Importantly, both genes were required for protection indicating that production of WonAB is both necessary and sufficient for anti-phage activity (Fig. 1f). Notably, although WonAB is unique to the

WASA-1 prophage found in *V. cholerae*, known defence systems could also be found at the same locus in other non-cholerae WASA-1 (Supplementary Table 3), consistent with the observation that prophages often carry anti-phage defence systems to increase the fitness of their bacterial hosts[30–32].

Bioinformatic analysis revealed that WonA is an ABC ATPase with a predicted core β-barrel-like ATPase fold characteristic of the ABC-ATPase clade[33]. Motifs required for ATP binding and hydrolysis (Walker A, Q-loop, ABC signature, Walker B, D-loop and H-loop) were readily identifiable (Fig. 1g and Extended Data Fig. 3a). Moreover, structural modelling confidently predicted that WonA forms a dimer, in which these motifs are positioned at the dimer interface to form the composite ATP-binding sites typical of ABC-ATPases[33] (Fig. 1g and Extended Data Figs. 3b and 4a,b). By contrast, WonB is predicted to encode a variant of the PD-(D/E)xK nuclease fold, with the highly conserved active site residues clustered together[34] (Fig. 1g and Extended Data Fig. 3c,d). Variants designed to disrupt either WonA ATP-binding and hydrolysis (K34A, R182A, E300A, H333A) or WonB nuclease activity (K71A) all abolished protection against ICP1, despite mostly being produced at similar levels to the wild-type (WT) control (Fig. 1h and Extended Data Fig. 3e). Interestingly, western blotting showed that inactivating WonA ATPase activity resulted in the loss of WonB, suggesting that WonA regulates or otherwise affects the stability of WonB (Extended Data Fig. 3e). Indeed, the two proteins are predicted to form a complex, with WonB binding to a surface created by the dimerized C-terminal extensions of WonA (Fig. 1g and Extended Data Fig. 4a–d).

Homologues of WonAB are found in diverse classes of mainly Gram-negative bacteria and overall were present in 0.43% of bacterial genomes examined (Extended Data Fig. 3f and Supplementary Table 4). Notably, when expressed in *Escherichia coli*, WonAB showed robust activity against all members of the Vequintavirinae subfamily of the Bacteriophage Selection for Your Laboratory (BASEL) collection[35], indicating that WonAB activity is not limited to ICP1 (Extended Data Fig. 3g). Interestingly, WonA ATPase motifs show deviations typical of the OLD family of ABC-ATPases[33,36] (Extended Data Fig. 3a). However, members of this family, such as OLD, Gabija, PARIS and the related Septu, are distinct from WonA in both structure and domain composition[36–41]. By contrast, WonAB appears to be closely related to an operon encoding a predicted OLD-ABC ATPase and a novel version of the restriction-enzyme fold (Extended Data Fig. 4), detected during a recent comprehensive in silico survey of ABC-ATPases[33]. Moreover, homologues of this operon are found in 0.96% of bacterial genomes and encompass all detected homologues of WonAB (Supplementary Table 5). Furthermore, WonAB also shows similarity to operons with a similar organization such as PD-T4-4 that have also recently been identified (Extended Data Fig. 4), albeit with variations in the Insert1/2 regions and C-terminal extensions of the ATPase, and in some cases a distinct nuclease[31,42]. Taken together, these data suggest that WonAB is part of a larger as yet uncharacterized family of ABC-ATPase sensors that are paired with nuclease effectors.

### ICP1 defence occurs via lysis-independent abortive infection

To gain insights into how WonAB mediates ICP1 defence, we monitored the growth kinetics of ICP1-infected cells at different multiplicities of infection (MOI), in the absence and presence of WASA-1 (Fig. 2a). This revealed that WonAB provides robust protection against ICP1 at a low MOI when only a subset of cells is initially infected. By contrast, infection at a high MOI, when most cells are simultaneously infected, resulted in a growth arrest (Fig. 2a). This phenotype is characteristic of abortive infection, wherein individual cell viability is sacrificed before the completion of phage replication to protect the surrounding bacterial population[43]. Indeed, time-lapse microscopy showed that while both WT and ΔWASA-1 cells swell upon infection, cells of the ΔWASA-1 go on to lyse within ~20 min, whereas WT cells remain in this

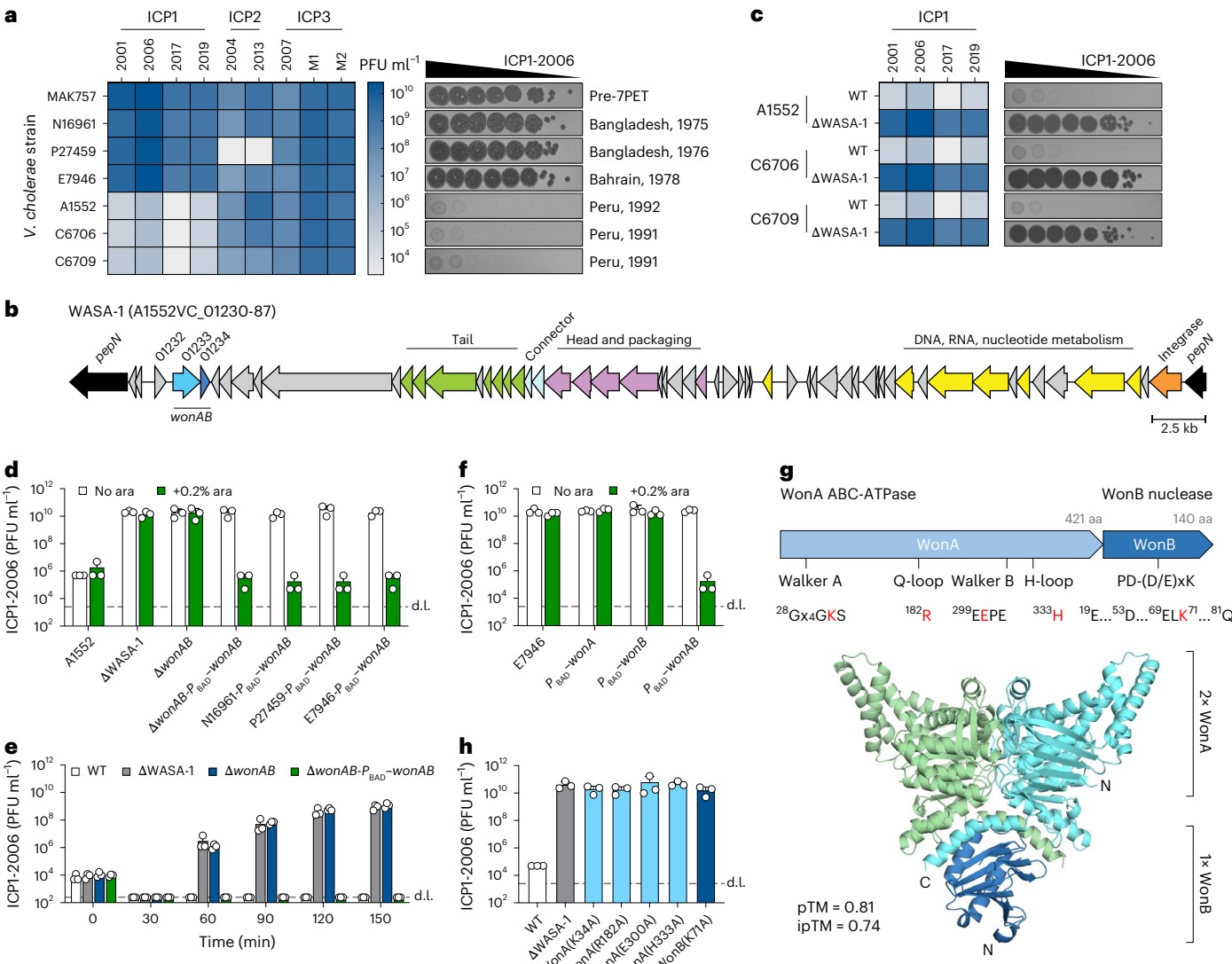

**Fig. 1 | WonAB carried on the WASA-1 prophage renders the WASA lineage ICP1 resistant. a**, Heat map showing mean plaque-forming units (PFU ml⁻¹) of diverse ICP1–3 isolates determined by plaque assays on wave 1 *V. cholerae* strains, compared with a pre-7th pandemic control (MAK757), shown alongside tenfold serial dilution plaque assays of ICP1-2006. Data represent the results of three independent experiments. Strain P27459 encodes a naturally occurring resistant variant of the ICP2 receptor OmpU. **b**, Schematic of the WASA-1 prophage highlighting *wonAB* and genes with predicted functions in prophage biology. **c**, Heat map showing mean PFU ml⁻¹ of diverse ICP1 isolates determined by plaque assays on Peruvian *V. cholerae* strains in WT and ΔWASA-1 backgrounds, determined as in **a**. **d**, Plaque assay showing the effects of ΔWASA-1, Δ*wonAB* and *wonAB* complementation in strain A1552 on ICP1-2006 PFU ml⁻¹ alongside the effect of producing WonAB in non-WASA strains. **e**, Replication assay showing

change in PFU ml⁻¹ over time following infection of the indicated A1552 cultures with approximately 10³–10⁴ PFU of ICP1-2006. **f**, Plaque assay showing the effect of E7946 derivatives producing either WonA, WonB or WonAB on ICP1-2006 PFU ml⁻¹. Where indicated, strains in **d**–**f** contain a chromosomally integrated transposon carrying the arabinose-inducible $P_{BAD}$ promoter, induced by the addition of 0.2% arabinose (ara). **g**, AlphaFold3-predicted structure of the putative WonAB complex shown below the schematic (top; aa, amino acids) indicating the conserved motifs. Residues targeted by site-directed mutagenesis are highlighted in red. N- and C-termini are indicated. **h**, Plaque assay testing the effect of WonA and WonB site-directed variants expressed from their native locus on ICP1-2006 PFU ml⁻¹. The bar charts show the mean + s.d. of three independent experiments. d.l., detection limit.

swollen state without lysing for the 60-min duration of the experiment (Fig. 2b, Supplementary Video 1 and Supplementary Fig. 1). Moreover, these non-growing cells persisted in an intact state for several hours before eventually undergoing lysis (Extended Data Fig. 5a,b). During this time, ICP1-infected cultures rebounded owing to the selection of spontaneous O1-antigen mutants (Extended Data Fig. 5c), as has been previously described[5]. Importantly, this is likely an in vitro phenomenon as O1-antigen mutants are attenuated for virulence in animal infection models[44].

Efforts to isolate either a spontaneous or an evolved ICP1 escape mutant able to overcome WonAB have so far proved unsuccessful.

Therefore, to study the mode of action of WonAB in more detail, we investigated different stages of the ICP1 life cycle in the absence and presence of WASA-1 (refs. 6,45). First, quantitative PCR (qPCR) showed that ICP1 DNA replication proceeds similarly in both strain backgrounds, albeit with a 2.7-fold defect in the WT background, indicating that WonAB neither restricts ICP1 DNA on entry nor prevents the initiation of DNA replication (Fig. 2c). Second, microscopy revealed that upon infection, the nucleoids of both strains undergo dramatic changes in morphology by 4 min postinfection, with the nucleoid frequently appearing to contract along its long axis, likely representing the action of ICP1-encoded nucleases and host-cell takeover[45,46] (Fig. 2d).

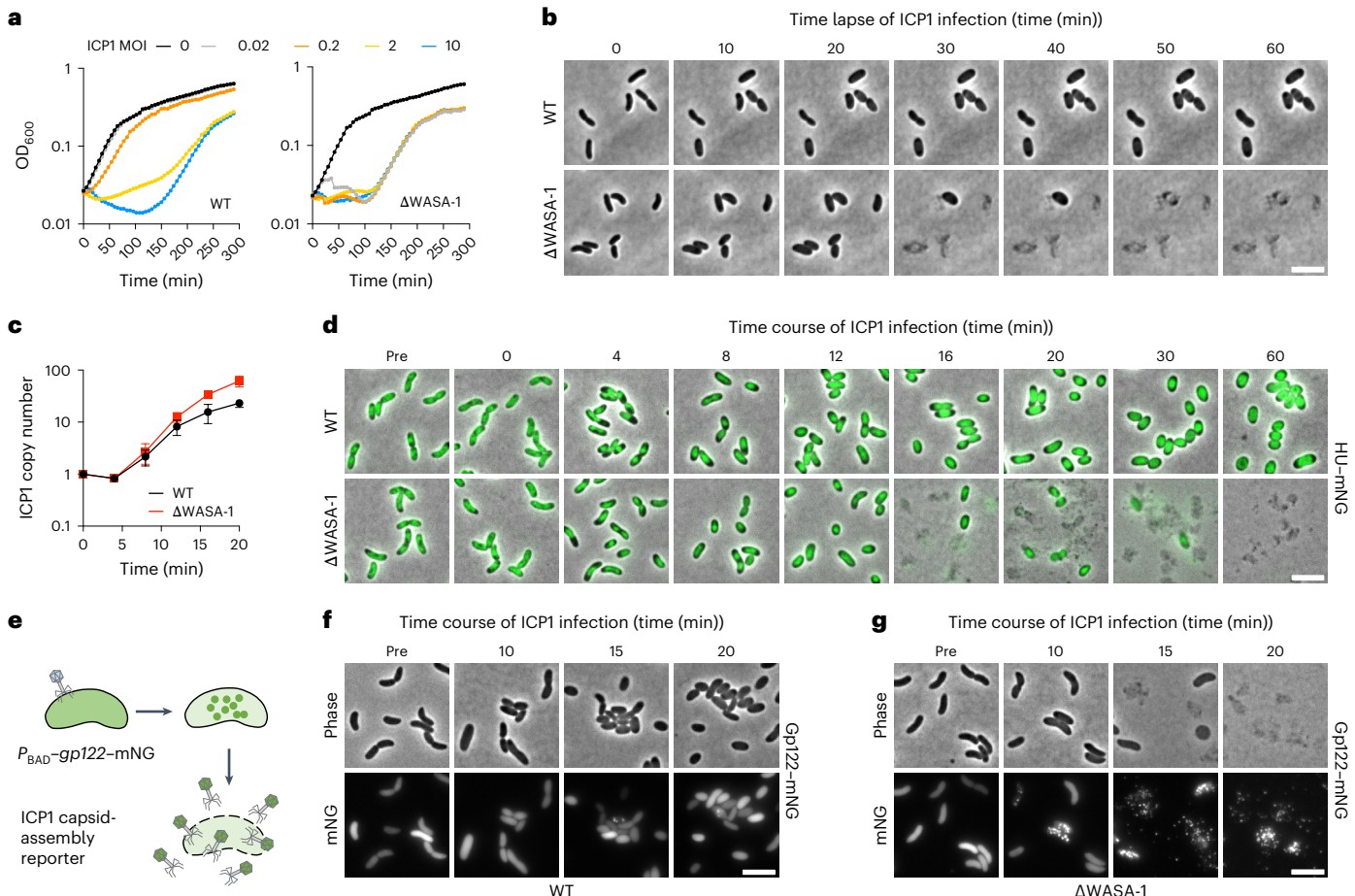

**Fig. 2 | WonAB protects cells by lysis-independent abortive infection.**
**a**, Growth kinetics of *V. cholerae* strain A1552 in the presence (WT) and absence of WASA-1 (ΔWASA-1), with either no phage or infected at time 0 with ICP1-2006 at the indicated MOI. Data are representative of three independent experiments. **b**, Time-lapse microscopy comparing exponentially growing cells of *V. cholerae* A1552 WT and ΔWASA-1 strains after infection with ICP1-2006 at MOI 5. **c**, Fold change in ICP1-2006 genome copy number relative to time 0, as determined by qPCR, after infection of either WT or ΔWASA-1 strains with ICP1-2006 at MOI 0.1. The chart shows the mean ± s.d. of three independent experiments.

**d**, Time-course microscopy snapshots comparing cell morphology and cellular DNA content, as monitored using HU–mNeonGreen fusion (mNG), in WT and ΔWASA-1 backgrounds, following infection with ICP1-2006 at MOI 5. **e–g**, Time-course microscopy snapshots comparing ICP1 capsid assembly, as monitored by the incorporation of a host-cell-produced mNG fusion to the ICP1 major capsid protein Gp122 (**e**), in either WT (**f**) or ΔWASA-1 (**g**) backgrounds, following infection with ICP1-2006 at MOI 5. All images are representative of the results of three independent experiments. Scale bars, 5 μm.

Notably, whereas ΔWASA-1 cells then rapidly went on to lyse, DNA in WT cells persisted in a highly compacted state and went on to form toroidal structures, suggesting that WonAB does not act by simply degrading DNA. Third, microscopy of ICP1-infected cells producing a fluorescent protein fusion to the ICP1 major capsid protein revealed robust capsid assembly, lysis and release in the ΔWASA-1 background (Fig. 2e–g). By contrast, capsid assembly was largely absent in WT cells (Fig. 2f). Taken together, these results show that WonAB does not prevent the initial damage inflicted by ICP1 and suggest that WonAB is activated after the initiation of ICP1 DNA replication and prevents progression to structural protein assembly.

**WonAB activation shuts down growth by inhibiting translation**
The data so far are consistent with WonAB aborting ICP1 infection via a lysis-independent mechanism that targets the host cell. To test whether WonAB can be artificially activated in the absence of infection, we over-expressed either the *wonAB* operon or each gene individually in an otherwise WT background (Fig. 3a and Extended Data Fig. 6a). While *wonAB* and *wonA* had no apparent phenotype, WonB overproduction was highly toxic, resulting in a growth arrest and a rapid loss in viability, with >99% of cells non-viable within 30 min (Fig. 3a–c and Extended Data Fig. 6a). Furthermore, toxicity was dependent on WonB nuclease

activity as the WonB(K71A) variant was non-toxic (Fig. 3b,c and Extended Data Fig. 6a). One explanation for these data could be that WonAB is a toxin–antitoxin system, with WonA acting as an antitoxin to neutralize the toxicity of WonB[47]. Arguing against this hypothesis, however, WonB overproduction was not toxic in a Δ*wonAB* background, despite being produced at similar levels to the control (Fig. 3a and Extended Data Fig. 6b,c). Moreover, deletion of *wonA* or inactivation of WonA ATPase activity also abolished the toxicity associated with WonB overproduction (Fig. 3a and Extended Data Fig. 6b,c). Thus, these results rule out a simple sequestration mechanism and suggest either that WonA is required to activate WonB or that the two proteins act together.

Microscopy revealed that following WonB overproduction, growth-arrested cells show condensed nucleoids that at later stages display aberrant morphologies and are highly compacted (Fig. 3d and Extended Data Fig. 6d). Notably, the vast majority of cells retained DNA and purified genomic DNA also remained intact (Extended Data Fig. 6d,e), indicating that toxicity does not result from chromosomal DNA degradation. As the nucleoid compaction phenotype resembled the effects of antibiotics that inhibit translation[48], we investigated WonB production over time using western blotting (Fig. 3e). Remarkably, this revealed that compared with the inactivated WonB(K71A) control, WonB is produced only for a short time after

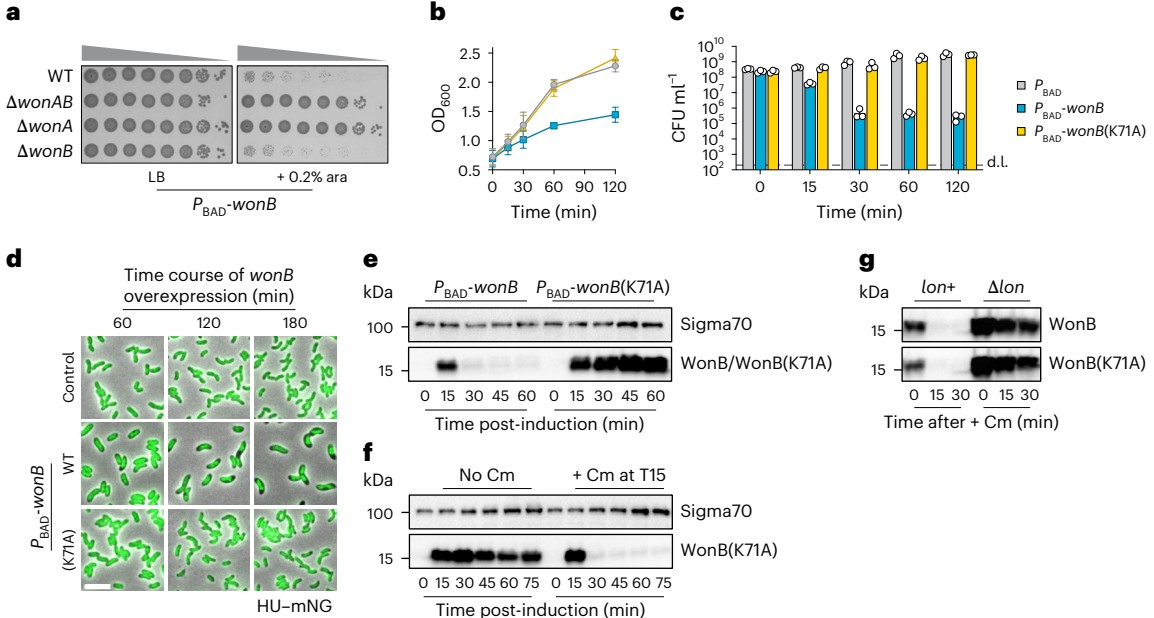

**Fig. 3 | WonAB activation shuts down cell growth by inhibiting translation.**
**a**, Toxicity assay evaluating the growth of tenfold serial dilutions of *V. cholerae* strain A1552 cultures and the indicated derivatives on solid media, in the absence (LB) and presence (+ 0.2% ara) of WonB overproduction. **b,c**, Toxicity assay evaluating the growth kinetics (**b**) and cell viability (**c**) of exponentially growing cultures in liquid media upon WonB overproduction following induction at time 0, as compared with a negative control strain and a strain overproducing the inactive WonB(K71A) variant. The charts show mean values. Error bars in **b** show ±s.d. and +s.d. in **c. d**, Time-course microscopy snapshots showing the effect of WonB overproduction on cell morphology and cellular DNA content, as monitored using HU–mNG fusion, compared with a negative control strain and a strain overproducing the inactive WonB(K71A) variant. Scale bar,

5 μm. **e**, Western blot showing protein levels of WonB and WonB(K71A) over time following induction in exponentially growing cells at time 0. Predicted molecular mass of WonB, 16.1 kDa. Sigma 70 was used as a loading control. **f**, Western blot showing protein levels of WonB(K71A) over time following induction in exponentially growing cells at time 0, in the absence and presence of chloramphenicol (Cm), added at 15 min (T15). **g**, Western blot showing protein levels of WonB and WonB(K71A) over time following Cm addition, in the presence (*lon+*) and absence of Lon protease (Δ*lon*). Overexpression of genes in **a–g** was done from a chromosomally integrated transposon carrying the arabinose-inducible $P_{BAD}$ promoter, induced by the addition of 0.2% arabinose. All images, charts and blots are representative of the results of three independent experiments.

induction and, by 30 min post-induction, was no longer detectable. Importantly, quantitative reverse-transcription PCR (qRT-PCR) showed that *wonB* mRNA levels were induced similarly between the two constructs and remained stable throughout the course of the experiment (Extended Data Fig. 6f). These data suggested that (1) WonB overproduction inhibits translation and (2) WonB is likely unstable. In agreement with these hypotheses, following inhibition of translation with chloramphenicol, WonB rapidly disappeared in a Lon protease-dependent manner (Fig. 3f,g), indicating that WonB is indeed unstable and subject to proteolytic degradation. As in the absence of ICP1 WonB was toxic only when produced in excess of WonA, Lon-mediated degradation likely functions as a safety mechanism to reduce WonB accumulation and hence prevent autoimmunity. Collectively, these results support a model whereby, upon activation, WonAB aborts phage infection by inhibiting translation, and thus prevents the progression of the ICP1 life cycle. Notably, although WonAB aborts the infection, cells are probably already irreversibly damaged by ICP1 during the initial stages of the infection. Furthermore, as the growth arrest upon WonB overproduction does not result in genomic DNA degradation, these results suggest that the target of WonB nuclease activity could be an rRNA or a tRNA.

## VSP-II^WASA carries two distinct anti-phage defence systems

The second genetic signature of the WASA lineage is a unique variant of VSP-II (Fig. 4a), wherein the chemotaxis gene cluster from the prototypical VSP-II is replaced by a set of three previously undescribed genes (A1552VC_00274-76) and a frame-shifted transposase gene[8,25,49]. Although these genes are not predicted to be part of a known defence system, and lacked activity against ICP1–3, bioinformatic analysis revealed that they encode proteins with domains characteristic of

defence systems (see below). Consistent with this, when this cluster was expressed in *E. coli*, we observed anti-phage activity against several major groups of the BASEL collection (Fig. 4b and Extended Data Fig. 7a). As outlined below, genetic dissection revealed that this cluster encodes two independent systems, with the two-gene operon VC_00274-75 encoding the GmrSD-like Type IV REase of WASA strains (GrwAB) (Fig. 4c–e) and VC_00276 encoding the *V. cholerae* Shedu (VcSduA) (Fig. 4f–h).

### GrwAB is a modification-dependent restriction system
GrwAB showed potent anti-phage activity against all members of the Queuovirinae and Tevenvirinae (Straboviridae), as well as some members of the Markadamsvirinae (Demerecviridae), and both genes were required for protection (Fig. 4b). Interestingly, the signature feature of the Queuovirinae and Tevenvirinae subfamilies is that their genomes contain hypermodified nucleobases with either 7-deazaguanine-modified guanosines or sugar-modified hydroxymethyl cytosines, respectively, suggesting that GrwAB may recognize DNA modification[35,50]. In line with this hypothesis, GrwAB resembles the GmrSD family of modification-dependent restriction enzymes, such as GmrSD, BrxU and SspE[51–54] (Fig. 4c and Extended Data Fig. 8a–d). These are typically single-protein systems composed of an N-terminal DUF262 domain that likely functions as a DNA modification sensor, and that uses NTP binding and hydrolysis to regulate the activity of a C-terminal His-Me nuclease domain (DUF1524), which functions as an effector to target non-self DNA[51,53,54]. Indeed, GrwA is predicted to share this architecture, with an N-terminal DUF262 domain containing the highly conserved QR, DGQQR and FxxxN motifs[51], coupled with an α-helical linker to a C-terminal DUF1524 domain containing

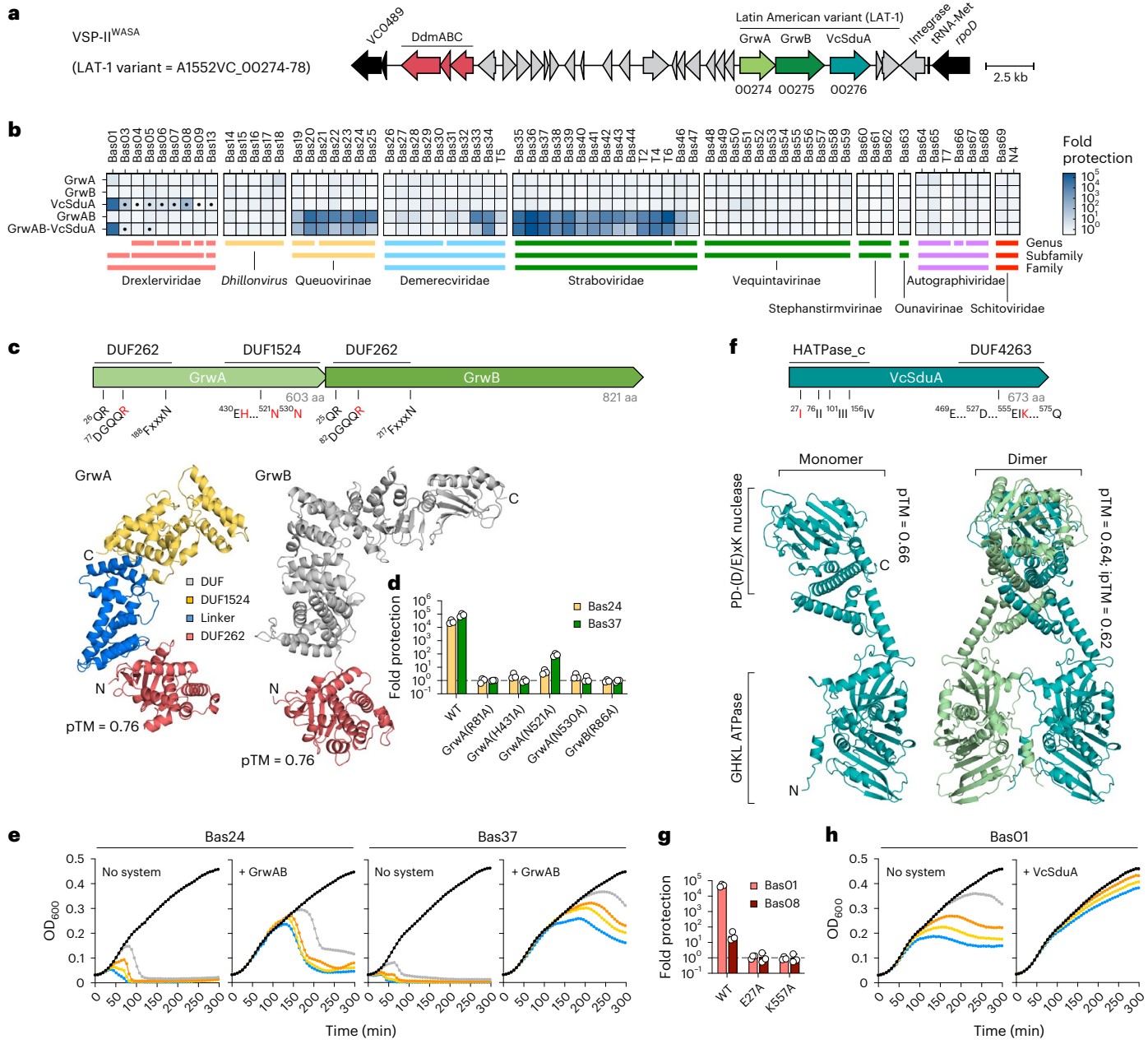

**Fig. 4 | VSP-II^WASA variant encodes two distinct anti-phage defence systems.**
**a**, Schematic of VSP-II^WASA highlighting the LAT-1 variant present in the WASA lineage and the previously characterized DdmABC system. **b**, Fold protection against *E. coli* phages of the BASEL collection conferred by the production of the indicated systems and components in *E. coli*, compared with a negative 'no system' control. The boxes with a filled dot indicate phages showing altered plaque morphology. Data represent the mean of two independent experiments. **c,f**, Predicted structures of GrwAB (**c**) and VcSduA (**f**) generated using AlphaFold3 shown below schematics (top) indicating the conserved domains and motifs identified in each system. Residues targeted by site-directed

mutagenesis are highlighted in red. **d,g**, Fold protection against the indicated *E. coli* phages conferred by site-directed variants of GrwAB (**d**) and VcSduA (**g**) in *E. coli*, compared with a negative 'no system' control. The bar charts show the mean + s.d. of three independent experiments. **e,h**, Growth kinetics of *E. coli* cultures in the absence (No system) and presence of either GrwAB (**e**) or VcSduA (**h**), either with no phage or infected at time 0 with the indicated MOI. Data are representative of three independent experiments. Systems were expressed from a chromosomally integrated transposon using the arabinose-inducible $P_{BAD}$-promoter. Growth media were supplemented with 0.2% arabinose.

the canonical ββα fold and catalytic residues of the His-Me nuclease superfamily[55] (Fig. 4c and Extended Data Fig. 8a–c). By contrast, and unlike known GmrSD systems, GrwB encoded by the second gene contains an N-terminal DUF262 domain followed by multiple domains of unknown function (Fig. 4c and Extended Data Fig. 8d). Notably, variants designed to disrupt either the NTP hydrolysis of DUF262 in GrwA(R81A) and GrwB(R86A), or the nuclease activity of DUF1524 in GrwA(H431A)/(N521A)/(N530A), all abolished anti-phage activity (Fig. 4d), indicating

that GrwA and GrwB function together. Indeed, structural modelling of a potential GrwAB complex supports the idea that the two proteins interact (Extended Data Fig. 8e).

Importantly, multiple independent escaper mutants of Bas24 were obtained that overcome defence by GrwAB (Extended Data Fig. 9a). These escapers all contain changes in either the active site or the DNA-binding surface of DpdA—the transglycosylase that inserts the 7-deazaguanine modification into the DNA—consistent with the

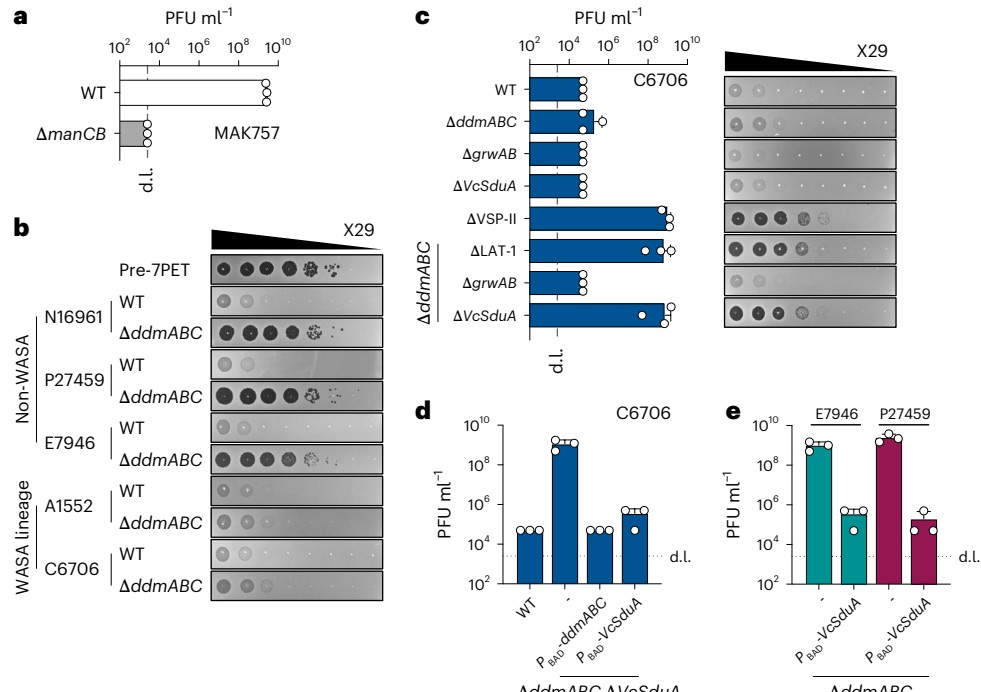

**Fig. 5 | VcSduA protects *V. cholerae* against vibriophage X29. a**, Plaque assay showing X29 PFU ml⁻¹ on the pre-7th pandemic strain MAK757, in the presence and absence (Δ*manCB*) of the O1-antigen. **b**, Tenfold serial dilution plaque assays of X29 on wave 1 *V. cholerae* strains, comparing the effect of deleting *ddmABC* in non-WASA (N16961, P27459, E7946) and WASA lineage (A1552, C6706) strains, below a pre-7th pandemic control (MAK757). **c**, Plaque assay showing X29 PFU ml⁻¹ obtained on C6706 derivates deleted for combinations of *ddmABC* with ΔLAT1 (Δ*grwAB-VcSduA*), Δ*grwAB* and Δ*VcSduA*, alongside representative images

of each tenfold serial dilution plaque assay. Note that strain C6706 was used in these assays because the rifampicin antibiotic resistance marker in strain A1552 partially interferes with X29 infection. **d,e**, Plaque assays testing the ability of VcSduA production to complement X29 protection in a C6706Δ*ddmABC*Δ*VcSduA* background (**d**) and in Δ*ddmABC* backgrounds of the non-WASA linage strains E7946 and P27459 (**e**). The bar charts show the mean + s.d. of three independent experiments. The images are representative of the results of three independent experiments.

disruption of DNA modification leading to immune evasion[56] (Extended Data Fig. 9b–d). Interestingly, growth kinetics of phage-infected cultures producing GrwAB lacked the growth arrest or premature lysis typical of abortive infection (Fig. 4e). However, cultures did go on to show moderate-to-severe lysis, which microscopy revealed was due to the presence of non-dividing filamentous cells (Supplementary Video 2 and Supplementary Fig. 2), suggesting that either GrwAB is unable to clear the infection before the cell sustains irreversible damage or that the cell is subject to collateral damage during phage defence. Finally, bioinformatic analyses revealed that homologues of GrwAB are found in ~1% of genomes examined (Extended Data Fig. 8f and Supplementary Table 6), suggesting that GrwAB belongs to an emerging family of two-gene GmrSD-like modification-dependent restriction enzymes that includes the recently discovered TgvAB system present in *V. cholerae* VPI-2 (refs. 22,23). Similar to TgvAB, the requirement for both GrwA and GrwB suggests that these proteins probably act as a complex, which could represent either a previously undescribed mode of action or else a mechanism to overcome phage-encoded anti-defences. Notably, despite the similarities, GrwAB has an expanded range compared with TgvAB and is also able to recognize 7-deazaguanine-modified DNA. Furthermore, although GrwA, TgvB and GmrSD all share a similar domain architecture, except for the shared N-terminal DUF262 domain, GrwB and TgvA are distinct from one another[22].

## VcSduA is a Shedu system with a GHKL ATPase N-terminal domain

VcSduA showed activity against all members of the T1-like Drexlerviridae family, with strong protection against Bas1 (>10,000-fold), Bas03 and Bas08 (10–100-fold), while the remainder showed a reduced plaque diameter that, in the case of Bas03 and Bas05, was drastic (Fig. 4b and

Extended Data Fig. 7b). Sequence analysis and structural modelling showed that VcSduA encodes an N-terminal GHKL (Gyrase, Hsp90, Histidine Kinase, MutL) ATPase domain (HATPase_c) containing the highly conserved motifs I–IV that define this ATPase superfamily[57,58], followed by a C-terminal PD-(D/E)xK nuclease domain (DUF4263) that is also found in the Shedu defence system[18,59,60] (Fig. 4f and Extended Data Fig. 10a–c). Variants designed to disrupt either ATP hydrolysis VcSduA(E27A) or nuclease activity VcSduA(K557A) both abolished anti-phage activity despite being produced at similar levels to WT control (Fig. 4g and Extended Data Fig. 10d). Furthermore, growth kinetic experiments with phage-infected cultures revealed similarly robust protection at all tested MOIs (Fig. 4h), consistent with VcSduA targeting and restricting the phage directly while preserving the viability of the host cell.

Proteins with a similar domain organization to VcSduA were previously annotated as paraMORC3 and have recently been proposed to be part of the Shedu defence system family owing to the presence of the signature Shedu domain DUF4263 (refs. 58,59). Indeed, Shedu has recently been reclassified as consisting of a shared nuclease domain with diverse N-terminal regulatory domains (type I), including various enzymatic domains (type I-D)[59]. Strikingly, modelling of VcSduA as a dimer (Fig. 4f), which is typical of GHKL ATPases, confidently predicted an entwined configuration with the GHKL ATPase domain sitting below a cavity formed by a clamp-like middle domain, which in other GHKL family members mediates binding to DNA and protein substrates[57,61]. We therefore hypothesize that the GHKL ATPase domain functions to control the activity of the C-terminal nuclease domain. Interestingly, modelling of VcSduA as a tetramer was not well supported (Extended Data Fig. 10e). Furthermore, VcSduA homologues were present in ~1.2% of genomes examined (Extended Data Fig. 10f

and Supplementary Tables 7 and 8), although GrwAB and VcSduA were rarely found together outside of *V. cholerae*, consistent with the demonstration above that they are independent systems. Overall, these data show that VcSduA is a previously undescribed member of the Shedu family and validates it as a member of the type I-D class.

### VcSduA protects *V. cholerae* against vibriophage X29

Next, to determine whether GrwAB and VcSduA were functional in their native *V. cholerae* host, we first searched the National Center for Biotechnology Information (NCBI) virus database for vibriophages with modified genomes that might be targeted by GrwAB (Methods). However, no such phage targeting *V. cholerae* was identified (Supplementary Table 9). We therefore screened a collection of phages from the Félix d'Hérelle Reference Center for Bacterial Viruses, focusing on phage X29 (ref. 62) as we determined that it requires the O-antigen for infection (Fig. 5a). Importantly, a previous study recently showed that the VSP-II-encoded DdmABC defence system[21] protects *V. cholerae* against X29 (ref. 63). Indeed, deletion of *ddmABC* in the non-WASA 7PET strains tested here was sufficient to fully restore X29 plaque formation (Fig. 5b). By contrast, deletion of *ddmABC* in the WASA lineage strains A1552 and C6706 had no effect (Fig. 5b), raising the possibility that the LAT-1-encoded defence systems of VSP-II[WASA] were active against X29. Consistent with this idea, while deletion of *grwAB* or *VcSduA* alone had no effect, deletion of either the entire LAT-1 region or *VcSduA* alone in a Δ*ddmABC* background restored X29 plaque formation to levels comparable to that of a complete VSP-II deletion (Fig. 5c). Furthermore, ectopic expression of *VcSduA* complemented the Δ*ddmABC*Δ*VcSduA* deletion and conferred complete protection against X29 in non-WASA strains deleted for *ddmABC* (Fig. 5d,e). Collectively, these data show that VcSduA is active in its native host organism at native expression levels and that WASA lineage strains have two distinct mechanisms to defend against phage X29.

## Discussion

The main discovery of this study is that the two unique genetic signatures (WASA-1 and VSP-II[WASA]) of the WASA lineage encode three previously undescribed antiviral defence systems: WonAB, GrwAB and VcSduA. Previously, these elements have primarily been used as a molecular signature to track the origins of these strains[8,25–27,64], but their importance for the success of the lineage and the Latin American cholera epidemic has remained unknown. Here we show that combined, these systems have broad activity against diverse bacteriophage families and target two-thirds of the isolates in the BASEL bacteriophage collection, including those with highly modified genomes. Importantly, we also show that in the native *V. cholerae* host, the WonAB system carried on the WASA-1 prophage renders strains resistant to ICP1—the major predatory phage of pandemic *V. cholerae*[6]. Given the ongoing arms race between *V. cholerae* and ICP1 (refs. 9,12) and the suggested role of ICP1 predation in cholera epidemics[3–5], it is therefore reasonable to assume that this would have provided a substantial selective advantage to the lineage. Nevertheless, as WonAB can also target other phages, then together with the systems carried on VSP-II[WASA], this selective advantage may have been against phages other than ICP1. Indeed, the VSP-II[WASA]-encoded VcSduA system protects WASA lineage strains against vibriophage X29, a myovirus that was originally isolated from a cholera patient in India in 1927[62]. Notably, spacers targeting X29 are abundant in the CRISPR–Cas systems present in classical 6th pandemic and non-7PET strains[65,66], suggesting that X29 (or related phages) may be commonly encountered. Interestingly, as VcSduA shares redundancy with DdmABC, it could potentially serve to protect against X29 variants that have evolved DdmABC resistance. However, given VcSduA activity against Drexlerviridae in *E. coli*, it is equally possible that it functions primarily against other non-DdmABC-sensitive phages.

Before the start of the epidemic in 1991, Latin America had been free from pandemic cholera for almost 100 years[24]. However, following the introduction of the WASA lineage from West Africa in the late 1980s, an explosive cholera epidemic spread rapidly throughout Latin America, with >500,000 cases in Peru alone in 1991–1992, and ongoing outbreaks continuing until 2001[8,24,27,28,67]. Given our findings, we therefore propose that the combined activities of the defence systems encoded on WASA-1 and VSP-II[WASA] directly contributed to the success of the WASA lineage and envisage at least two possible scenarios by which this could have occurred. Firstly, the acquisition of these systems in Africa could have allowed the strains to resist local vibriophages, triggering the proliferation of the lineage that therefore facilitated the transfer to South America. Notably, cholera outbreaks were ongoing in Western and Central Africa just before the start of the Latin American epidemic, and intensified in the years afterwards[27,28]. Furthermore, strains containing WASA-1 and VSP-II[WASA] continued to circulate in Western and Central Africa up until the late 1990s, including strains responsible for the 1987–1996 Angolan cholera epidemic[27,28,68]. Remarkably, we found WASA-1 (99.7% identity to A1552) in a recently sequenced non-7PET *V. cholerae* strain isolated in Switzerland in 2019 from a patient with a travel history to Morocco[69], suggesting that WonAB-containing WASA-1 continues to circulate in Africa.

Secondly, the defences provided by WASA-1 and VSP-II[WASA] could have played a more direct role in the transmission, for example, by allowing the WASA lineage to overcome an entry barrier posed by phages associated with local non-pandemic *V. cholerae* strains endemic to South America that might have prevented previous introductions. In support of these proposals, the ability to defend against phage predation is thought to play a key role in the observed evolution of local *V. cholerae* lineages, which then go on to be successful before subsequently being replaced[9,12]. Indeed, the continuous antagonistic co-evolution of bacterial defences and phage-encoded anti-defences drives the gain and loss of defence systems in natural *Vibrio* populations, with such defences often encoded on mobile genetic elements and accounting for a large proportion of the accessory genome[16,70]. Thus, while 7PET strains are typically considered to be clonal based on the phylogeny of the core genome, closely related strains can harbour diverse mobile genetic elements encoding defence systems that vary over time in response to the dominant phages[9,12]. This, together with the results presented here, reinforces the need to consider the role of the accessory genomes in the success of newly emerging lineages and therefore understand their pandemic potential. Finally, while further work will be necessary to decipher the mechanistic details of the defence systems discovered here, this study provides fresh insights into the evolution of one of the major epidemic lineages of the seventh cholera pandemic.

## Methods

### Bacterial strains, plasmids and bacteriophages

The bacterial strains, plasmids, oligonucleotides, bacteriophages and defence systems used in this study are detailed in Supplementary Tables 10–16. The primary *V. cholerae* strain used throughout this study, A1552, is an O1 El Tor (Inaba) strain isolated from a patient in 1992 following an outbreak of cholera linked to a commercial airline flight between Lima, Peru, and Los Angeles, California, United States[71,72]. The primary *E. coli* strain used throughout this study is the non-arabinose-metabolizing strain MG1655Δ*araCBAD* (ref. 73). *E. coli* strains S17-1λ*pir* and MFD*pir* were used for cloning and bacterial mating with plasmids containing the conditional R6K origin of replication[74,75].

### Growth conditions

Bacteria were cultured either in lysogeny broth (LB (Miller); 10 g l⁻¹ tryptone, 5 g l⁻¹ yeast extract, 10 g l⁻¹ NaCl (Carl Roth)) with shaking at 180 rpm or on LB + 1.5% agar plates, at 37 °C. Transformation of *V. cholerae* was done on chitin powder (Alfa Aesar via Thermo Fisher) suspended in half-concentrated Instant Ocean medium (Aquarium Systems) and sterilized by autoclaving. Thiosulfate citrate bile salts sucrose

agar plates were used for counter-selection against *E. coli* following bacterial mating (Sigma-Aldrich). SacB-based counter-selection for allelic exchange was done on NaCl-free media containing 10% sucrose. Counter-selection of *pheS*\*-carrying strains (see below) was done on LB agar plates supplemented with 20 mM 4-chloro-phenylalanine. Ampicillin (100 μg ml$^{-1}$), chloramphenicol (2.5 μg ml$^{-1}$), gentamicin (25–50 μg ml$^{-1}$), kanamycin (75 μg ml$^{-1}$), streptomycin (100 μg ml$^{-1}$) and rifampicin (100 μg ml$^{-1}$) were used for selection, as required. Growth of MFD*pir* was supported by the addition of 0.3 mM diaminopimelic acid (Sigma-Aldrich). Unless stated otherwise, genes under the control of the $P_{BAD}$ promoter were induced by supplementing growth media with 0.2% L-arabinose.

### Strain construction

Genetic engineering of *V. cholerae* was done using natural competence for transformation to introduce deletions marked with antibiotic resistance genes, which when necessary were removed using flippase (FLP)-based recombination (TransFLP[76]). Alternatively, antibiotic cassettes linked to the modified phenylalanyl-tRNA synthetase *pheS*\* (*pheS*(A294G/T251A)) were removed by natural transformation using the Trans2 method[77]. Scarless, markerless modifications were introduced using allelic exchange with derivatives of the counter-selectable plasmid pGP704-Sac28 (ref. [78]). A mini-Tn7 transposon carrying *araC* and the indicated gene(s) of interest under control of the $P_{BAD}$ promoter was integrated downstream of the *glmS* locus in *V. cholerae* and *E. coli* by introducing derivatives of pGP704-TnAraC via triparental mating, as previously described[79]. Plasmids were constructed using either standard restriction enzyme cloning or inverse PCR, and were used exclusively as intermediates for strain construction. All strains and constructs were verified by PCR, and either Sanger sequencing or Oxford Nanopore Technologies-based full plasmid sequencing (Microsynth AG). Software used for data analysis was SnapGene v.4.3.11 and Geneious Prime v.11.0.14.1.

### *V. cholerae* bacteriophage methods

Vibriophages were propagated on *V. cholerae* strains E7946 or MAK757 (ICP1 and ICP3) and E7946Δ*manCB* (ICP2). Briefly, overnight cultures were back diluted and grown in LB at 37 °C, 180 rpm to exponential phase (optical density at 600 nm (OD$_{600}$), approximately 0.5–0.6), infected with the relevant phage at a MOI of <0.01. Following lysis of the culture, debris was removed by centrifugation (10 min, 4,000 × *g*, room temperature (RT)) and lysates were passed through a 0.2-μm filter and stored at 4 °C after the addition of chloroform to 1%. Phage titres were determined by plaque assays on strain E7946 using the small-drop plaque assay on double-layer plates, as described below. To quantify ICP and X29 phage infections, 100 μl and 10 μl, respectively, of overnight culture were added to molten top agar (LB + 0.5% agar), poured on top of a bottom agar (LB + 1.5% agar) and allowed to solidify at RT. For complementation of *wonAB* and *VcSduA*, cultures were grown either for 2 h + 0.2% arabinose or overnight with 0.02% arabinose, respectively, and the top agar was also supplemented with arabinose at the respective concentration. Phages were serially diluted ($10^{-1}$–$10^{-8}$), and 4 μl of each dilution was spotted onto plates and allowed to dry under a laminar-flow hood. Plates were imaged after overnight incubation at 37 °C, and the PFU ml$^{-1}$ enumerated. The theoretical detection limit in this experiment was $2.5 × 10^3$ PFU ml$^{-1}$. In cases in which no discrete plaques were observed but a zone of non-specific lysis was apparent (for example, ICP1 on strain A1552; Fig. 1a), the last dilution showing this phenotype was counted as 20 plaques. To determine whether these zones of non-specific lysis resulted from phage propagation or 'lysis from without' (that is, lysis at high phage concentration without viable phage production) re-streak tests were performed, as previously described[16]. Material from the highest phage concentration spot on plaque assay plates was re-streaked onto freshly poured top agar plates containing the ICP1-sensitive strain A1552ΔWASA-1.

### *E. coli* bacteriophage methods

The BASEL collection of *E. coli* bacteriophages was obtained from A. Harms (ETH, Zürich)[35]. Infections were quantified using the small-drop plaque assay described above, with the following modifications. Briefly, overnight cultures of *E. coli* MG1655Δ*araCBAD* were back diluted 1:100 in 3 ml LB + 5 mM CaCl$_2$, 20 mM MgSO$_4$ and 0.2% arabinose and grown for 2 h at 37 °C, 180 rpm to reach exponential phase, before being diluted 1:40 in molten top agar, supplemented with 5 mM CaCl$_2$, 20 mM MgSO$_4$ and 0.2% arabinose. Phages were serially diluted ($10^{-1}$–$10^{-8}$), 5 μl of each dilution spotted onto plates, and allowed to dry under a laminar-flow hood. Plates were imaged after overnight incubation at 37 °C, and the PFU ml$^{-1}$ enumerated. The theoretical detection limit in this experiment was $2 × 10^3$ PFU ml$^{-1}$. Fold protection was calculated as the ratio of PFU ml$^{-1}$ on the strain of interest to that of the no-system control strain *E. coli* MG1655Δ*araCBAD*-TnAraC.

### Quantification of infectious ICP1 phages

To monitor the production of infectious ICP1 phages, overnight cultures of the *V. cholerae* strains indicated in the text were back diluted 1:100 in 2.5 ml LB in 14-ml test tubes and grown at 37 °C, 180 rpm to exponential phase (OD$_{600}$ approximately 0.5–0.6) before being infected with approximately $10^3$–$10^4$ PFU ml$^{-1}$ of ICP1-2006. Infected cultures were incubated at 37 °C, 180 rpm, and sampled at 30-min intervals for 2.5 h. At each time point, 300 μl of culture was mixed with 30 μl of chloroform, vortexed vigorously for 10 s and centrifuged (3 min; 20,000 × *g*; RT). To determine the starting titre at time 0, samples were prepared from LB-only controls. Chloroform-free supernatants were transferred to new tubes and serially diluted ($10^0$–$10^{-7}$), and the PFU ml$^{-1}$ were enumerated through a plaque assay on plates seeded with the ICP1-sensitive *V. cholerae* strain E7946. The theoretical detection limit in this experiment was $2.5 × 10^2$ PFU ml$^{-1}$.

### Bacteriophage infection kinetics

To monitor the kinetics of ICP1 infection, overnight cultures of the *V. cholerae* strains indicated in the text were back diluted 1:100 in 2.5 ml LB in 14-ml test tubes and grown at 37 °C, 180 rpm for 1 h and 45 min to exponential phase (OD$_{600}$ approximately 0.5–0.6). Subsequently, 180-μl aliquots of cultures diluted 1:5 in pre-warmed LB were added to the wells of a 96-well plate containing 20 μl of ICP1-2006 at the indicated MOI (in technical triplicate). Bacterial growth (OD$_{600}$) at 37 °C with shaking was followed using a SpectraMax i3x (Molecular Devices) plate reader at 6-min intervals for a total of 49 cycles. The kinetics of Bas01, Bas24 and Bas37 infection in *E. coli* were monitored exactly as described above, except that *E. coli* strains were grown for 2 h in LB + 5 mM CaCl$_2$, 20 mM MgSO$_4$ and 0.2% arabinose and aliquots were diluted 1:10. *E. coli* phages used for plate reader experiments were propagated on MG1655Δ*araCBAD*, as described above.

### Microscopy

Imaging was conducted using Zeiss Axio Imager M2 and Zeiss Axio Observer Z1 epifluorescence microscopes, equipped with AxioCam MRm cameras and controlled by Zeiss Zen software (v.2.6 blue edition). Images were captured using a Plan-Apochromat 100×/1.4-NA Ph3 oil objective, and for fluorescence microscopy, illumination was provided by an HXP120 lamp. For snap-shot imaging, cells were immobilized on slides coated with 1.2% (wt/vol) agarose in phosphate-buffered saline, covered with a number 1 coverslip. For time-lapse microscopy, cells were immobilized on slides coated with 1.2% (wt/vol) agarose in either LB (ICP1 infection of *V. cholerae*) or LB + 5 mM CaCl$_2$, 20 mM MgSO$_4$ and 0.2% arabinose (Bas24 infection of *E. coli*) and imaged automatically at the indicated time points, within a temperature-controlled stage-top chamber (H301-K-Frame; OKOLAB) set to 37 °C with an accompanying objective heater. Images were prepared for publication using ImageJ (version 2.1.0/1.53h; imagej.net/software/fiji), and where needed, drift was corrected using the HyperStackRegPlus function of MicrobeJ[80].

## Time-lapse and time-course microscopy of ICP1 infection

To image ICP1 infection in *V. cholerae*, overnight cultures of the strains indicated in the text were back diluted 1:100 in 2.5 ml LB in 14-ml test tubes and grown at 37 °C, 180 rpm for 1 h and 45 min to exponential phase (OD$_{600}$ approximately 0.5–0.6). For time-lapse microscopy, an aliquot of the culture was then mixed with ICP1-2006 to achieve an MOI of 5 and immediately transferred to a slide and imaged, as described above. For time-course microscopy, 180-μl aliquots of cultures were mixed with 20 μl ICP1-2006 in 2-ml microcentrifuge tubes to achieve an MOI of 5 and incubated flat at 37 °C, 180 rpm. At the indicated time points, tubes were removed from the incubator and samples immediately transferred to a slide and imaged, as described above. To monitor cellular DNA content, strains were used encoding a C-terminal mNeonGreen fusion to the histone-like DNA-binding protein HU (HU–mNeonGreen) separated by a GSGSGS linker, and expressed from the native *hupA* locus. To monitor ICP1 capsid assembly, infections were performed in strains with a chromosomally integrated transposon carrying a C-terminal mNeonGreen fusion to the ICP1 major capsid protein Gp122 (Gp122–mNeonGreen), separated by a GSGSGS linker and expressed under the control of the arabinose-inducible $P_{BAD}$ promoter. The fusion was induced by the inclusion of 0.02% arabinose in the growth media.

## Time-lapse microscopy of Bas24 infection

To image Bas24 infection in *E. coli*, overnight cultures were back diluted 1:100 in 3 ml LB + 5 mM CaCl$_2$, 20 mM MgSO$_4$ and 0.2% arabinose in 14-ml test tubes and grown at 37 °C, 180 rpm for 2 h to mirror the conditions used for the BASEL screen above. An aliquot of the culture was then mixed with Bas24 to achieve an MOI of 10 and allowed to stand at RT for 2 min before being transferred to a slide and imaged.

## qPCR of ICP1 genome replication

ICP1 genome replication was quantified by qPCR using a protocol modified from ref. [11]. Briefly, overnight cultures were back diluted 1:100 in 2.5 ml LB in 14-ml test tubes and grown at 37 °C, 180 rpm for 1 h and 45 min to exponential phase (OD$_{600}$ approximately 0.5–0.6). ICP1-2006 was then added to an MOI of 0.1, tubes were briefly vortexed to ensure proper mixing and a 20-μl sample immediately withdrawn (time ($t$) = 0) and heat inactivated at 95 °C for 20 min. Sampling was repeated following incubation at 37 °C, 180 rpm for 4, 8, 12, 16 and 20 min postinfection. Samples were analysed using a LightCycler 96 Instrument (Roche) with SYBR Green I (Fast Start Essential DNA Green Master; Roche) to detect amplification products using the ICP1-specific primers qICP12006E_148-f (ACTTTGGTGCGTGAA-GAAGG) and qICP12006E_148-r (ACTTGCTCACCTGAATGGTC). ICP1 genome copy number levels are presented relative to $t$ = 0 and were analysed with LightCycler 96 software v.1.1.0.1320 (Roche) with the absolute quantification method using a standard curve of purified ICP1-2006 genomic DNA.

## WonB toxicity assay

Overnight cultures were back diluted to OD$_{600}$ 0.05 in LB and grown for 1 h and 45 min at 37 °C, 180 rpm to exponential phase (OD$_{600}$ approximately 0.5–0.6), induced by the addition of 0.2% arabinose ($t$ = 0), and incubation continued for 120 min. The OD$_{600}$ of each culture was measured at $t$ = 0, 15, 30, 60 and 120 min. At each time point, samples were collected and processed to determine the colony-forming units (CFU) ml$^{-1}$ by serial dilution and plating on LB agar, and transcript levels via qRT-PCR, and genomic DNA was prepared using a GenElute Bacterial Genomic DNA Kit (NA2110; Sigma-Aldrich) according to the manufacturer's instructions. Samples to determine protein levels by western blotting were collected at the time points indicated in the text and processed as described below. To determine the stability of WonB protein levels over time, translation was inhibited by the addition of chloramphenicol to 200 μg ml$^{-1}$.

## qRT-PCR of *wonAB* transcript levels

From the WonB toxicity assay (above), 2-ml culture samples were collected by centrifugation (3 min; 20,000 × $g$; 4 °C), and cell pellets were resuspended in Tri-Reagent (Sigma-Aldrich), snap-frozen in a dry ice–ethanol bath and stored at −80 °C. RNA extraction, cDNA synthesis and qPCR were then performed exactly as previously described[81], using a LightCycler 96 Instrument (Roche). Transcript levels are presented relative to mRNA levels of the reference gene *gyrA*, and were analysed with LightCycler 96 software v.1.1.0.1320 (Roche) using the standard curve method. Primers used to detect *wonA* and *wonB* were as follows: qVC_01233-F (GTCCACAAGCAAACGGTAAG) and qVC_01233-R (TTCTC-CCATGAATACCTCGG); qVC_01234-F (ACAACTGTCTCAAAGGATCC) and qVC_01234-R (GCTCAATATCCAGTCACATC).

## Western blotting

Unless stated otherwise, overnight cultures were back diluted 1:100 and grown at 37 °C, 180 rpm for 3 h. Lysates were prepared by resuspending pelleted bacteria in 2× Laemmli buffer (Sigma-Aldrich) and normalized to optical density (100 μl buffer per OD unit), and samples were then heated at 95 °C, 15 min. Proteins were resolved on Mini-PROTEAN TGX Stain-Free precast gels (10% or 12%, as required; Bio-Rad) and transferred onto PVDF membranes using a Trans-Blot Turbo Transfer System (Bio-Rad) according to the manufacturer's instructions. Membranes were blocked in 2.5% skim milk in TBST (1× Tris-buffered saline with 0.1% Tween-20) with agitation at either RT for 1 h or at 4 °C overnight. Primary antibodies were added at a dilution of 1:500 in TBST, and membranes incubated at RT for 1 h. Membranes were then washed three times with TBST before being incubated at RT for 1 h with anti-rabbit IgG conjugated to HRP (A9169, Sigma-Aldrich) diluted 1:20,000 in TBST. Membranes were washed as above and visualized by chemiluminescence using Lumi-Light$^{PLUS}$ Western Blotting Substrate (Roche). Primary antibodies against WonA (2210455), WonB (2210453) and VcSduA (2310059) were custom raised in rabbits against synthetic peptides (Eurogentec) and their specificity validated by comparing strains lacking the relevant proteins. Uniform sample loading was verified by the intensity of non-specific bands on the membranes or by detection of Sigma 70 (1:20,000 diluted Direct-Blot HRP anti-*E. coli* RNA Sigma 70 Antibody, 663205, BioLegend).

## Identification of defence systems, protein domains, motifs and structural predictions

The repertoire of known defence systems present in *V. cholerae* strains was identified using DefenseFinder (v.1.2.4) and PADLOC (v.2.0.0) (accessed 14 May 2024); see Supplementary Table 1 for details[82,83]. Conserved protein domains were initially identified by searching the PFAM and NCBI Conserved Domain Database with the MOTIF Search tool (genome.jp/tools/motif/; accessed 22 July 2024), with the search supplemented by remote homology searches using HHpred (PDB_mmCIF70_8_Mar, default database; accessed 22 July 2024), and structural alignments were performed with DALI using structural predictions generated by AlphaFold3 (refs. [84–86]). The predicted template modelling (pTM) score and, where appropriate, the interface pTM (ipTM) score, are shown alongside each model[86]. For full confidence metrics for AlphaFold models, see Supplementary Fig. 3. Sequence logos showing the conservation of motifs in each system were generated using Weblogo 3 from multiple sequence alignments done using MAFFT (multiple alignment using fast Fourier transform) within Jalview v.2.11.4.0 (refs. [87,88]).

## Identification and distribution of WASA-1

A BLASTN (basic local alignment search tool (BLAST) using standard nucleotide databases) search of the *V. cholerae* A1552 WASA-1 sequence (default parameters; *e*-value $1 \times 10^{-10}$) against 45,725 bacterial genomes (NCBI, RefSeq database, complete and chromosome-level genomes, downloaded on 22 May 2024) revealed hits exclusively in the *Vibrio*

genus. The BLASTN search was therefore repeated against a database consisting of 7,624 genomes, covering all available assemblies within the *Vibrio* genus (RefSeq database, genomes from contig to complete level assemblies, downloaded on 30 May 2024). A second complementary BLASTN search was also performed with the additional max_hsps (high-scoring pair) parameter set to 1. Initial hits were then manually curated to identify complete WASA-1 assemblies, as shown in Supplementary Table 2. WASA-1 conservation was visualized using clinker (v.0.0.28, default parameters)[89].

### Identification and distribution of defence systems

The presence of the WonAB, OLD-ABC ATPase + Novel REase, GrwAB and VcSduA defence systems was determined against 45,725 bacterial genomes (NCBI, RefSeq database, complete and chromosome-level genomes, downloaded on 22 May 2024) using MacSyFinder (v.2.1.1; default parameters)[90]. To build hidden Markov model (HMM) profiles, a position-specific iterated (PSI)-BLAST of each component was performed against the NCBI non-redundant protein sequence database (accessed in February 2023 and May 2024). The PSI-BLAST $e$-value threshold was set to $1 \times 10^{-10}$ and run for three iterations, except for VcSduA for which only one iteration was performed owing to NCBI computational constraints. Sequences were aligned using MAFFT (v.7.508; −−maxiterate 1000 −localpair parameters for alignment with higher accuracy) and HMM profiles built using the hmmbuild function of the HMMER suite (v.3.3.2; default parameters)[88,91]. To evaluate the distribution of each system in the RefSeq database used, genera represented by more than 500 genomes were extracted, and the NCBI common tree tool used to create an order-level phylogenetic tree showing the most represented clades, with the presence and absence of each system mapped. For the full list of identified hits, see Supplementary Tables 4−8.

### Isolation and characterization of Bas24 escaper phages that overcome GrwAB

Candidate escaper phages of Bas24, which appeared as spontaneous large plaques in the lowest dilution of the small-drop plaque assay on strains producing GrwAB, were isolated and propagated as follows. Single plaques of each candidate were resuspended in 90 μl phage buffer (50 mM Tris−HCL (pH 7.5), 100 mM MgCl$_2$, 10 mM NaCl), added to 1 ml exponentially growing *E. coli* MG1655Δ*araCBAD* producing GrwAB in a 14-ml test tube and cultured at 37 °C, 180 rpm for 3 h, at which point a further 2 ml of exponentially growing culture was added and incubated for 3 h, before phages were purified exactly as described above. As a control, a single plaque from the no-system control plate was picked and propagated in the absence of GrwAB production. Genomic DNA was prepared using the Norgen Phage DNA Isolation Kit (product number 46850, Norgen) according to the manufacturer's instructions. Whole-genome sequencing was done using the Small, 200-Mbp Illumina short read service on the Illumina NextSeq2000 platform (SeqCoast Genomics). Genomes were assembled using Unicycler (v.0.5.0)[92], using short reads only as input with the remaining parameters set to default, and assemblies rotated to match the Bas24 reference genome (GenBank: MZ501104). Genomes were aligned using MAFFT (v.7.508) and single-nucleotide polymorphisms called using snp-sites (v.2.5.1)[88,93].

### Search for DpdA-encoding vibriophages

To identify previously described vibriophages with modified genomes, phage sequences were downloaded from the NCBI virus database, with Vibrionaceae (taxid: 641) specified as the host. Using the resulting 1,700 nucleotide assemblies and the NCBI tool suite (v.2.14.1) to translate the sequences, a local database containing 117,034 protein sequences was generated (makeblastdb). This database was then searched for homologues of DpdA from phage Bas24 using Blastp ($e$-value $1 \times 10^{-10}$, rest of parameters to default). The resulting data are provided in Supplementary Table 9.

### Genome sequencing of Swiss WASA-1-carrying strain N19-2759

*V. cholerae* strain N19-2759 is a non-O1 and non-O139 non-toxigenic travel-associated clinical isolate taken from a patient in Switzerland in 2019 (Illumina-based sequencing accession number GCA_032819675.1)[69]. To obtain an assembled complete genome of strain N19-2759, genomic DNA was prepared as previously described[94], sequenced using Oxford Nanopore Technologies-based sequencing and assembled using the software flye (v. 2.9.3; Microsynth).

### Reporting summary

Further information on research design is available in the Nature Portfolio Reporting Summary linked to this article.

### Data availability

The genome assemblies of *V. cholerae* strain N19-2759 have been deposited in the NCBI GenBank database under accession number GCA_046097525.1. The raw reads are available from the Sequence Read Archive under submission number SRX26909066. The publicly available datasets used in this study and the dates that they were accessed are listed in Methods. All other data are available in the Article and Supplementary Information. Source data are provided with this paper.

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

## Acknowledgements

We thank all members of the Blokesch laboratory for constructive feedback throughout the project. We also gratefully acknowledge K. D. Seed (University of California, Berkeley), A. Camilli (Tufts University School of Medicine, Boston), W.-L. Ng (Tufts University School of Medicine, Boston) and A. Ali (University of Florida, Gainesville) for sharing ICP phages, and A. Harms (ETH, Zurich) for sharing the BASEL collection and for valuable scientific discussions. We thank the Félix

d'Hérelle Reference Center for Bacterial Viruses from the Université Laval (Québec, Canada) for providing phage X29 (HER66). Finally, we acknowledge R. Stephan and M. Biggel (University of Zurich) for sharing *V. cholerae* isolates from Switzerland and for valuable scientific discussions. This study was supported by grants from the European Research Council (724630) and the Swiss National Science Foundation (310030_185022), an HHMI International Scholarship (55008726) and EPFL intramural funding awarded to M.B.

## Author contributions

Conceptualization: D.W.A., M.J. and M.B. Methodology: D.W.A., M.J., A.L. and M.B. Investigation: D.W.A., M.J., A.L., S.S., L.R., L.B. and M.B. Visualization: D.W.A. Funding acquisition: M.B. Supervision: D.W.A. and M.B. Writing—original draft: D.W.A. Writing—review and editing: D.W.A., M.J. and M.B.

## Funding

## Competing interests

The authors declare no competing interests.

## Additional information

**Extended data** is available for this paper at https://doi.org/10.1038/s41564-025-02004-9.

**Correspondence and requests for materials** should be addressed to David W. Adams or Melanie Blokesch.

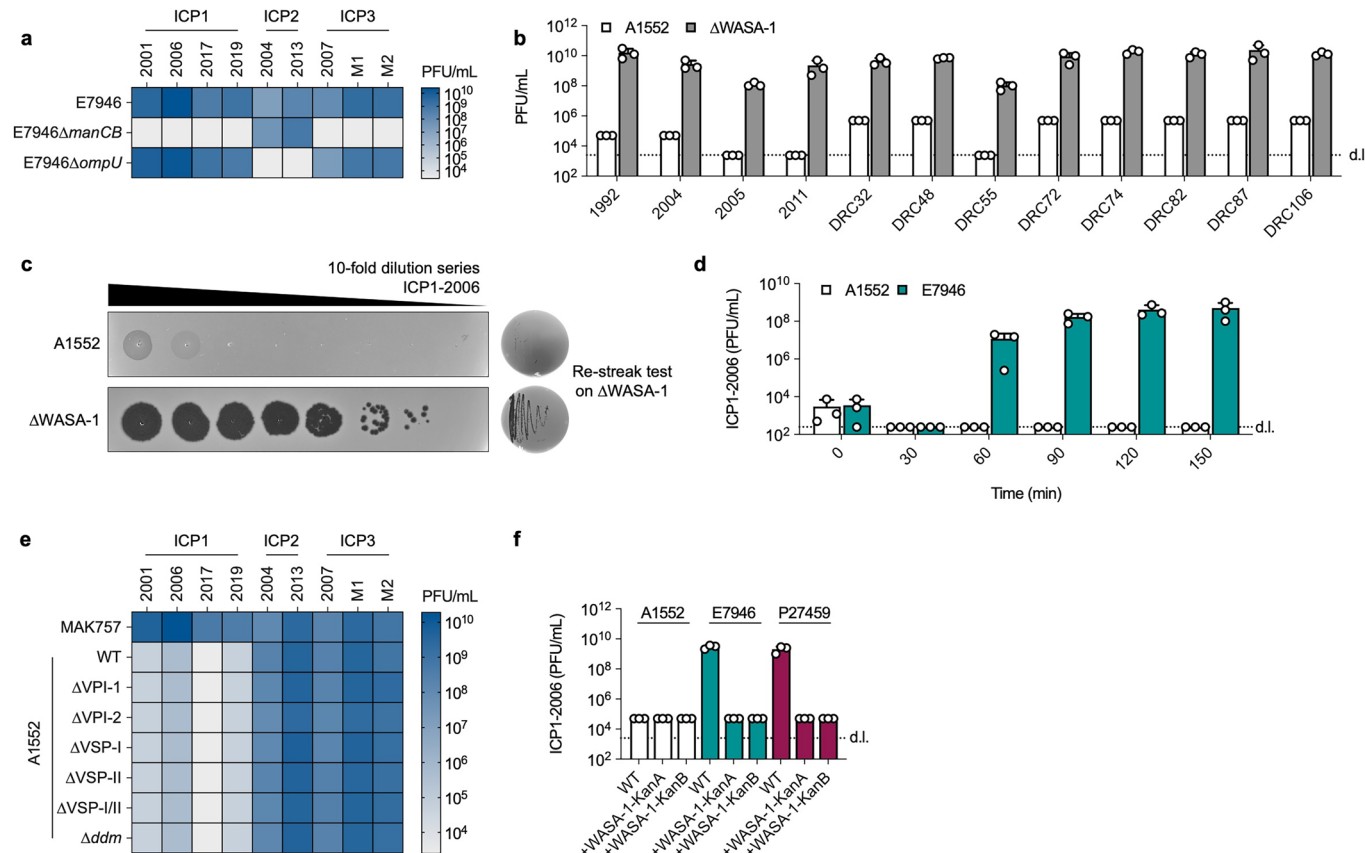

**Extended Data Fig. 1 | Control experiments showing role of WASA-1 prophage in ICP1 defence. (a)** Heat-map showing mean plaque-forming units (PFU/mL) of diverse ICP1-3 isolates determined by plaque assays on E7946 derivatives that lack either the O1 antigen receptor for ICP1 and ICP3 (E7946Δ*manCB*) or the OmpU receptor for ICP2 (E7946Δ*ompU*). **(b)** Plaque assay showing PFU/mL of various ICP1 isolates on A1552 in the presence and absence of WASA-1. **(c)** Re-streak tests evaluating the propagation of viable ICP1 phage progeny were conducted by taking material from the highest concentration spot of 10-fold serial dilution plaque assays done in the presence (A1552) and absence (ΔWASA-1) of WASA-1 (left) and re-streaking on fresh lawns of the susceptible ΔWASA-1 strain (right). **(d)** Replication assay showing change in PFU/mL over time following

the infection of A1552 and E7946 cultures with *c.a.* $10^3$-$10^4$ PFU of ICP1-2006. **(e)** Heat-map showing mean plaque-forming units (PFU/mL) of diverse ICP1-3 isolates determined by plaque assays on A1552 derivatives that lack the indicated genomic islands, as compared to the pre-7th pandemic control strain MAK757. **(f)** Plaque assay showing the effects of introducing WASA-1, using versions with a kanamycin resistance cassette inserted at two different positions, into the non-WASA lineage strains E7946 and P27459 on ICP1-2006 PFU/mL, as compared to the equivalent A1552 control strains. Heat-maps and bar charts show the mean from three independent experiments. Error bars show s.d. and d.l. = detection limit.

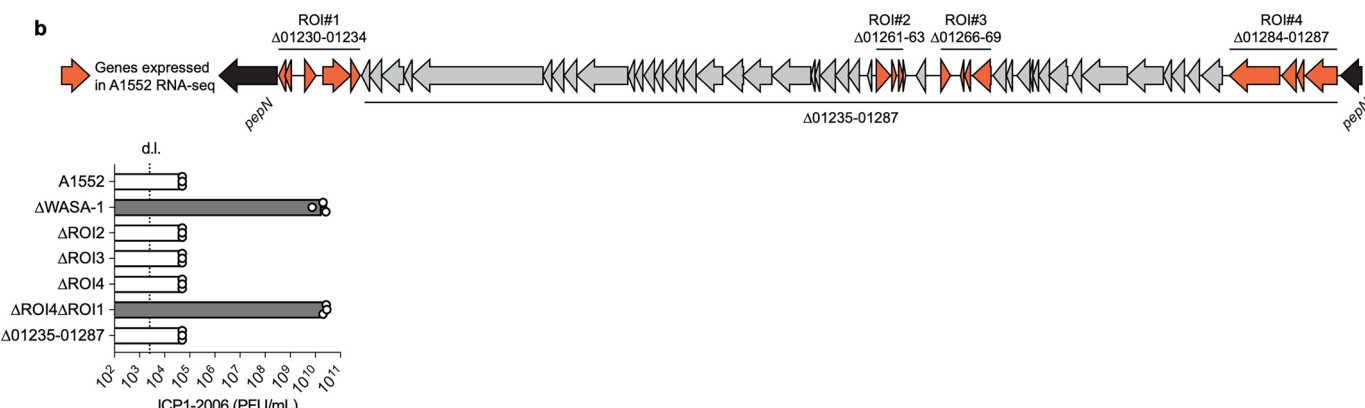

**Extended Data Fig. 2 | Conservation of the WASA-1 genome and identification of regions of interest.** (**a**) Schematic comparing the genome organisation of WASA-1 from *V. cholerae* strain A1552 with a representative set of examples of the WASA-1 detected by BLAST in diverse *Vibrio spp*. The schematic was built using the clinker pipeline and the connections between genomes are coloured according to protein identity (above the minimum default threshold of 30%). Genes with predicted functions in prophage biology are coloured according to function, as indicated. For details of WASA-1 identification and a full list of identified sequences, see methods and Supplementary Table 2. (**b**) Schematic of *V. cholerae* strain A1552 WASA-1 highlighting the genes that were highly expressed under standard growth conditions in RNA-seq results, and which were used to define the indicated regions of interest (ROI) 1-4. Note that ROI1 could only be deleted in the ΔROI4 background. The bar chart shows the effects of deleting each ROI on ICP1-2006 PFU/mL, as compared to WT and ΔWASA-1 control strains. Bars show the mean + s.d. from three independent experiments. d.l. = detection limit.

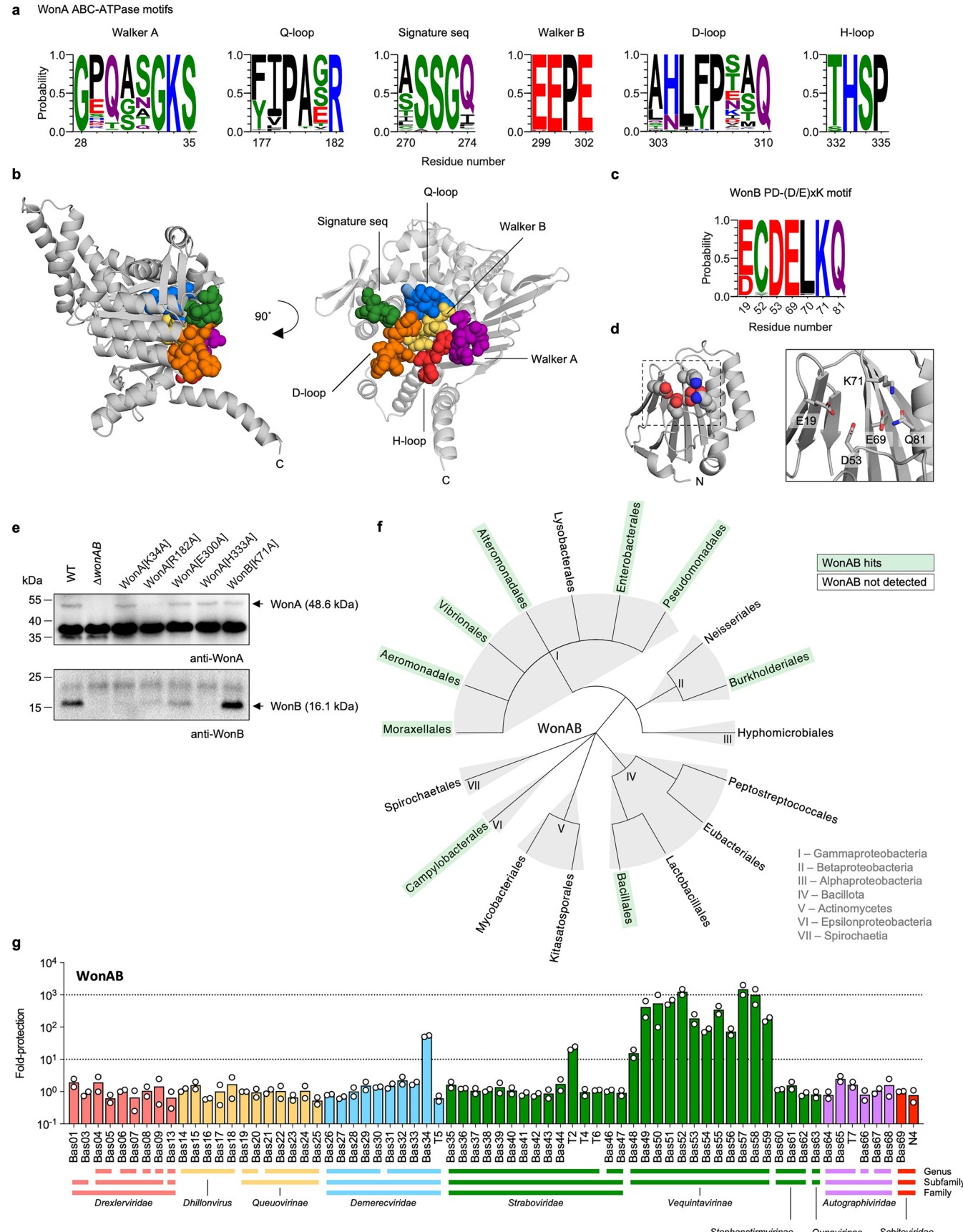

**a** WonA ABC-ATPase motifs

**c** WonB PD-(D/E)xK motif

**e**

**f**

WonAB hits

WonAB not detected

I – Gammaproteobacteria
II – Betaproteobacteria
III – Alphaproteobacteria
IV – Bacillota
V – Actinomycetes
VI – Epsilonproteobacteria
VII – Spirochaetia

**g**

**Extended Data Fig. 3 | See next page for caption.**

**Extended Data Fig. 3 | Identification of conserved motifs and distribution of WonAB.** (**a**) Sequence logos showing conservation of the identified ABC-ATPase motifs in the 198 WonA hits detected using MacSyFinder v.2.1.1, as compared to the equivalent residue number in *V. cholerae* A1552 WonA. Amino acids in logos are coloured according to chemical properties: polar (G, S, T, Y, C), green; neutral (Q, N), purple; basic (K, R, H), blue; acidic (D, E), red; and hydrophobic (A, V, L, I, P, W, F, M), black. (**b**) Location of the conserved ABC-ATPase motifs shown in (**a**) at the dimer interface of the WonA dimer structural prediction. (**c**) Sequence logo showing the conservation of the identified PD-(D/E)xK nuclease motif in the 198 WonB hits detected using MacSyFinder v.2.1.1, compared to the equivalent residue number in *V. cholerae* A1552 WonB. (**d**) Location of the identified PD-(D/E)xK nuclease residues in the WonB structural prediction. (**e**) Western blot

showing the protein levels of WonA and WonB, natively expressed in strain A1552 (WT) and in derivatives encoding the indicated site-directed variants. Blots are representative of the results of three independent experiments. (**f**) Distribution of WonAB hits detected using MacSyFinder v.2.1.1. The tree shows the order-level phylogeny of genera in the RefSeq database with more than 500 genomes (see methods). For the full list of 198 WonAB hits see Supplementary Table 4. (**g**) Anti-phage activity of WonAB in *E. coli*. Fold-protection against *E. coli* phages of the BASEL collection conferred by the production of WonAB in *E. coli* MG1655Δ*araCBAD*, as compared to a negative 'no system' control. The system was expressed from a chromosomally integrated transposon carrying the arabinose-inducible $P_{BAD}$- promoter, induced by the addition of 0.2% arabinose. Bar chart shows the mean of two independent experiments.

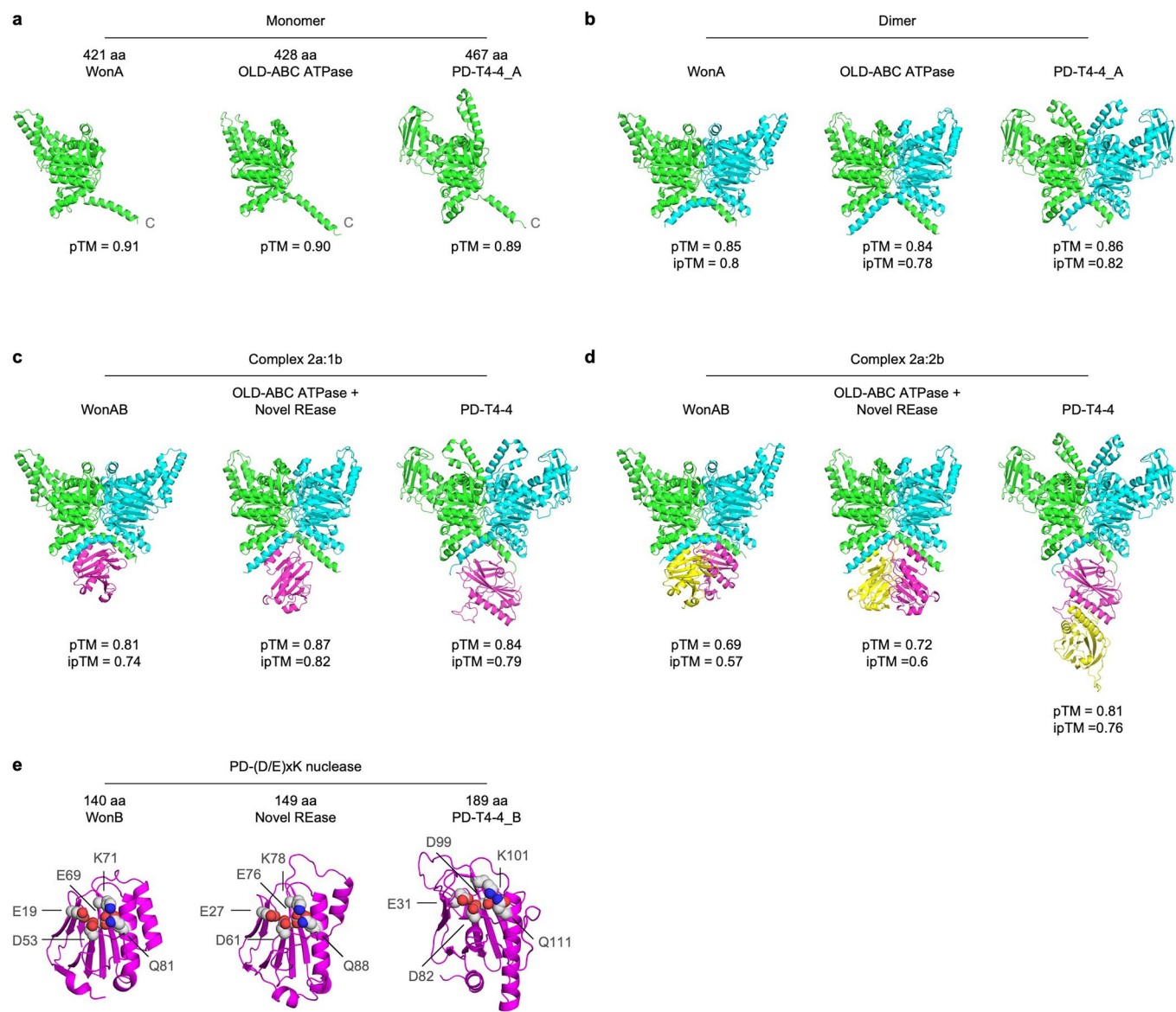

**Extended Data Fig. 4 | Summary of structural modelling of WonAB and related systems.** (**a**–**e**) Cartoon representations coloured by chain showing the AlphaFold3 predicted structures for WonAB, and the related OLD-ABC ATPase + Novel REase and PD-T4-4 systems. The ATPase components WonA, OLD-ABC ATPase and PD-T4-4_A were modelled as a monomer (**a**) and a dimer (**b**), and further modelled as a complex with either one (**c**) or two (**d**) copies of the nuclease components WonB, Novel REase and PD-T4-4_B. (**e**) Nuclease components modelled alone, highlighting the location of the predicted PD-(D/E) xK nuclease residues. The predicted template modelling (pTM) score, and where appropriate, the interface predicted template modelling (ipTM) score, are shown alongside each model. Protein sequences were obtained from NCBI: *V. cholerae* A1552 WonA (AWB73975.1) and WonB (AWB73976.1); *Anaerovibrio lipolyticus* DSM 3074 OLD-ABC ATPase (SHI83489.1) + Novel REase (SHI83462.1); *E. coli* MOD1-ECOR58 PD-T4-4_A (RCO57999.1) and PD-T4-4_B (RCO57988.1).

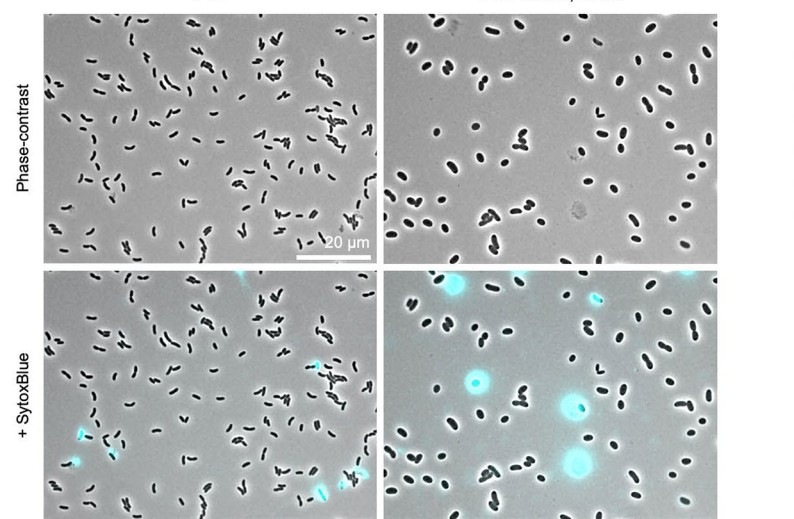

**Extended Data Fig. 5 | Imaging the fate of ICP1 infected cells. (a)** Time-course microscopy snapshots comparing the growth and cell morphology of *V. cholerae* A1552 cultures at 1, 2, 3, 4 and 5 hours after infection with ICP1-2006 at MOI 5, as compared to an uninfected control culture (+ LB). Note how the ICP1-infected culture initially contains only a few cells with normal morphology (white arrowheads), and that these cells increase in number as the culture rebounds, while the swollen ICP1-infected cells that are initially dominant persist for several hours before undergoing lysis (yellow arrowheads). **(b)** SytoxBlue-staining of *V. cholerae* A1552 cultures 1 hour after infection with ICP1-2006 at MOI 5, as compared to an uninfected control culture (+ LB). **(c)** Plaque assays testing the ability of ICP1-2006 and ICP3-2007 (which both require the O1 antigen receptor) to form plaques on cultures of A1552ΔWASA-1 that rebounded following ICP1-infection, as compared to a culture of the same strain that was grown without selection, and the set of indicated control strains. Note how neither ICP1-2006 nor ICP3-2007 are able to form plaques on the rebounded cultures, consistent with recovery of the culture being driven by the growth of spontaneous O1-antigen mutant cells. All images are representative of the results of three independent experiments. Scale bars = 20 μm.

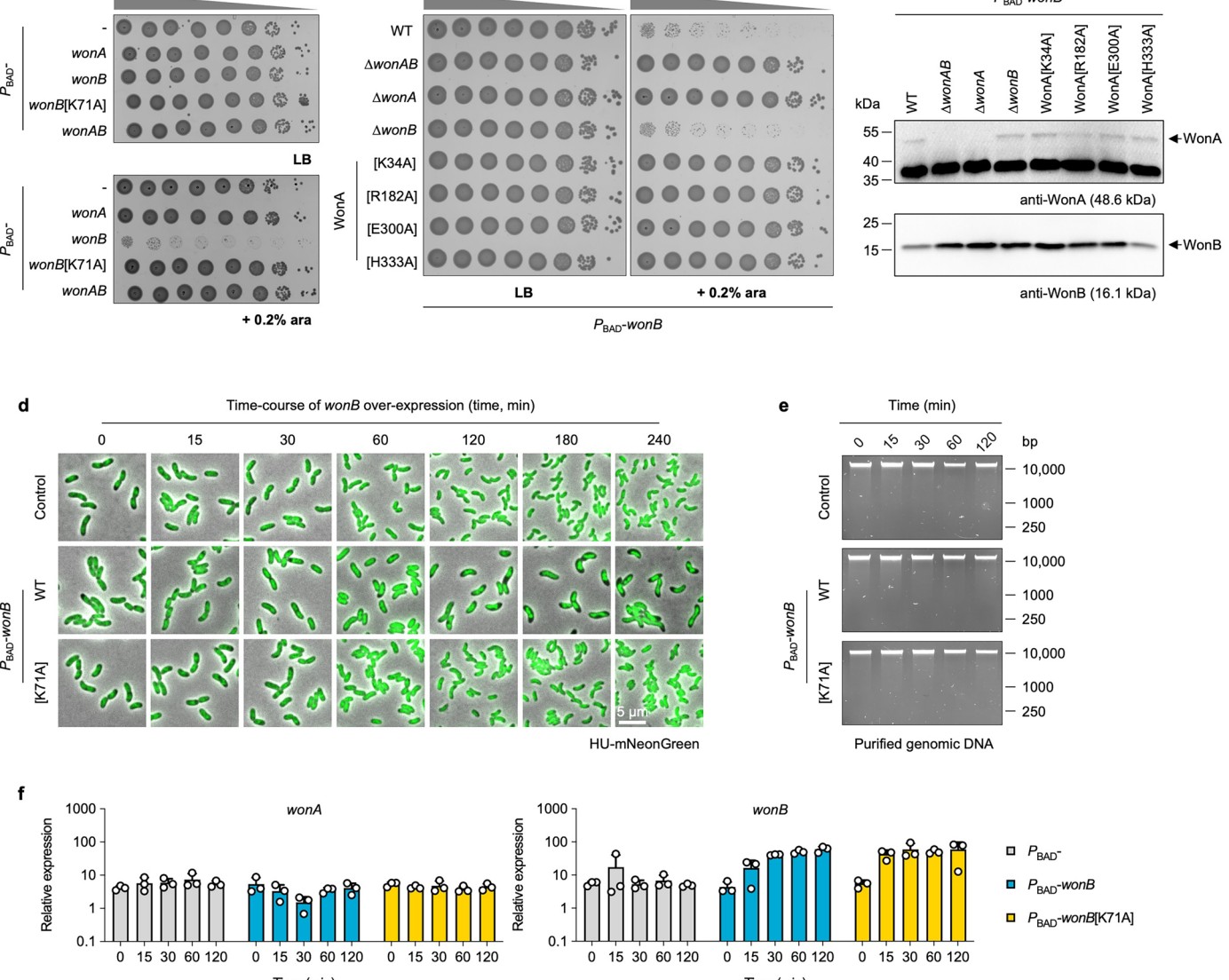

**Extended Data Fig. 6 | Control experiments showing details of toxicity during WonB overproduction.** (**a**) Toxicity assay evaluating the growth of 10-fold serial dilutions of cultures of *V. cholerae* strain A1552 derivatives with a chromosomally integrated transposon carrying the indicated genes under the control of the arabinose-inducible $P_{BAD}$- promoter, in the absence (LB) and presence of induction (+ 0.2% ara), as compared to a negative control strain. (**b**) Toxicity assay evaluating the growth of 10-fold serial dilutions of *V. cholerae* strain A1552 and the indicated derivatives encoding site-directed variants of WonA, compared to the indicated deletion control strains, in the absence (LB) and presence (+ 0.2% ara) of WonB overproduction. Note that the control strains are also shown separately in Fig. 3a. (**c**) Western blots showing the protein levels of WonA and WonB in the strains used for the toxicity assay shown in (**b**). Samples

were taken from exponentially growing cultures, 15 minutes after induction of $P_{BAD}$-*wonB*. (**d**) Time-course microscopy snapshots showing the effect of WonB overproduction on cell morphology and cellular DNA content, as monitored by a HU-mNeonGreen fusion, compared to a negative control strain and a strain overproducing the inactive WonB[K71A] variant. Scale bar = 5 μm. The panels depict the full time-series for the examples presented in Fig. 3d. (**e**, **f**) Agarose gels evaluating the integrity of genomic DNA extractions (**e**) and comparison of *wonA* and *wonB* transcript levels as determined by qRT-PCR (**f**), prepared from cultures over time upon WonB overproduction following induction in exponentially growing cells at time 0, as compared to a negative control strain and a strain overproducing the inactive WonB[K71A] variant. All data are representative of the results of three independent experiments. Bar charts show the mean + s.d.

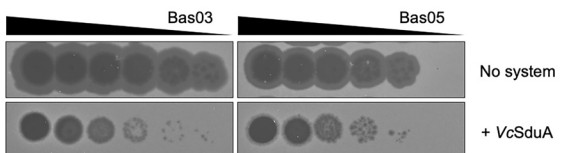

**Extended Data Fig. 7 | Anti-phage activity of GrwAB and *Vc*SduA in *E. coli*.**
(**a**) Fold-protection against *E. coli* phages of the BASEL collection conferred by
the production of either GrwA, GrwB, *Vc*SduA, GrwAB or GrwAB-*Vc*SduA in *E. coli*
MG1655∆*araCBAD*, as compared to a negative 'no system' control. Genes were
expressed from a chromosomally integrated transposon carrying the arabinose-
inducible $P_{BAD}$- promoter, induced by the addition of 0.2% arabinose. Bar charts
show the mean of two independent experiments. Bars with a black dot indicate
phages exhibiting altered plaque morphology. (**b**) Representative examples of
the altered plaque morphology phenotype observed for Bas03 and Bas05 in the
presence of *Vc*SduA production.

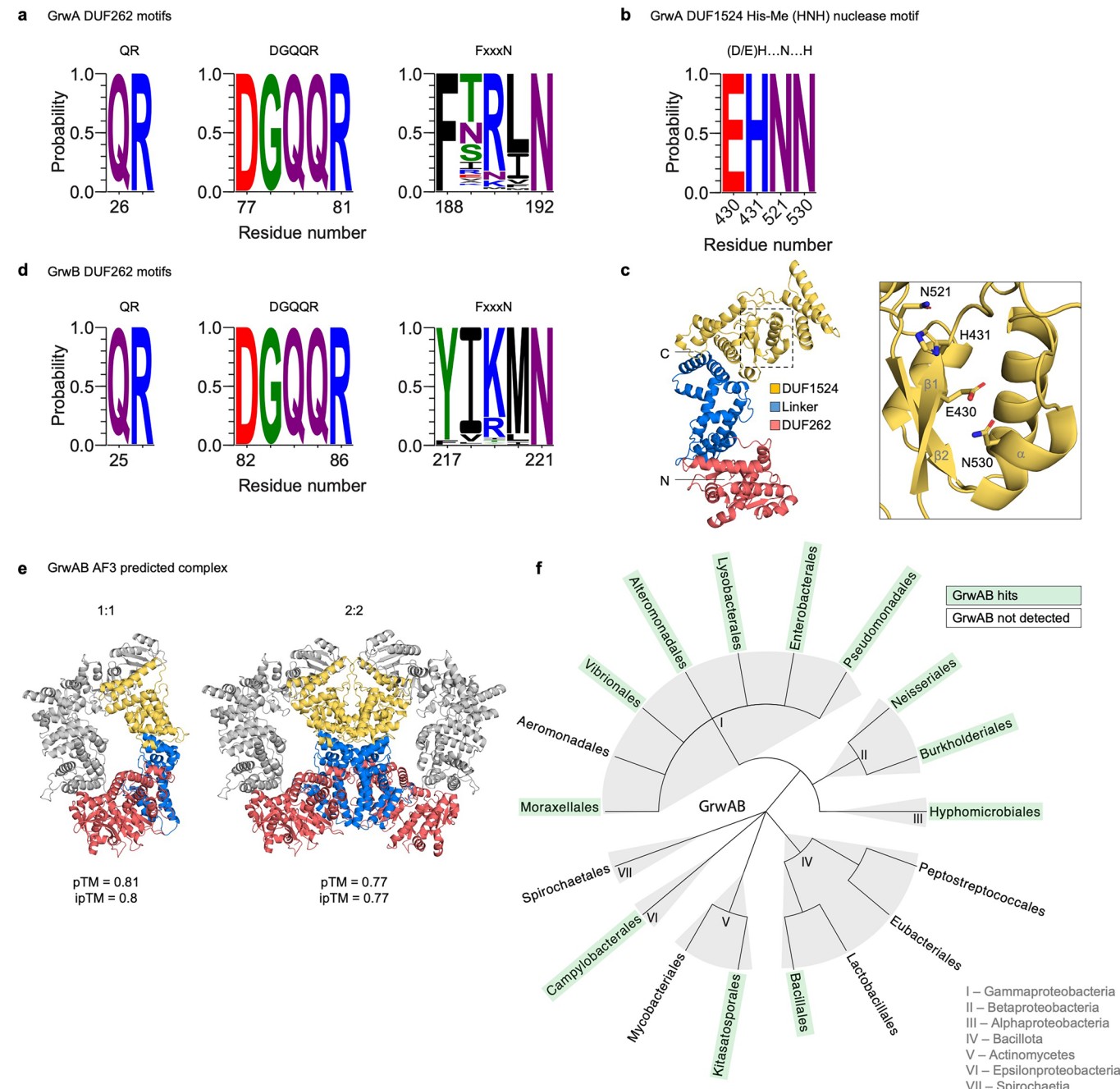

**Extended Data Fig. 8 | Identification of conserved motifs and distribution of GrwAB.** (**a**, **d**) Sequence logos showing conservation of the identified QR, DGQQR and FxxxN motifs in the DUF262 domains of the 471 GrwA (**a**) and 471 GrwB (**d**) hits detected using MacSyFinder v.2.1.1, as compared to the equivalent residue number in *V. cholerae* A1552 GrwA and GrwB. Amino acids in logos are coloured according to chemical properties: polar (G, S, T, Y, C), green; neutral (Q, N), purple; basic (K, R, H), blue; acidic (D, E), red; and hydrophobic (A, V, L, I, P, W, F, M), black. (**b**) Sequence logo showing the conservation of the identified His-Me (HNH) nuclease motif in the DUF1524 domain of the 471 GrwA hits detected

using MacSyFinder v.2.1.1, as compared to the equivalent residue number in *V. cholerae* A1552 GrwA. (**c**) Location of the identified His-Me (HNH) nuclease motif residues in the GrwA structural prediction, highlighting the predicted ββα fold. (**e**) AlphaFold3 predicted structures of potential GrwAB complexes. Colour scheme as in (**c**). (**f**) Distribution of GrwAB hits detected using MacSyFinder v.2.1.1. The tree shows the order-level phylogeny of genera in the RefSeq database with more than 500 genomes (see methods). For the full list of 471 GrwAB hits see Supplementary Table 6.

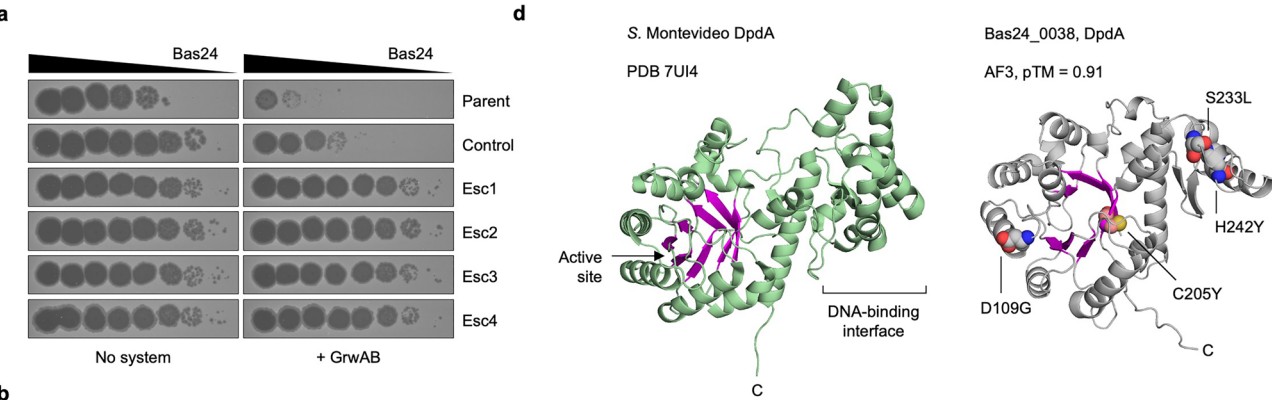

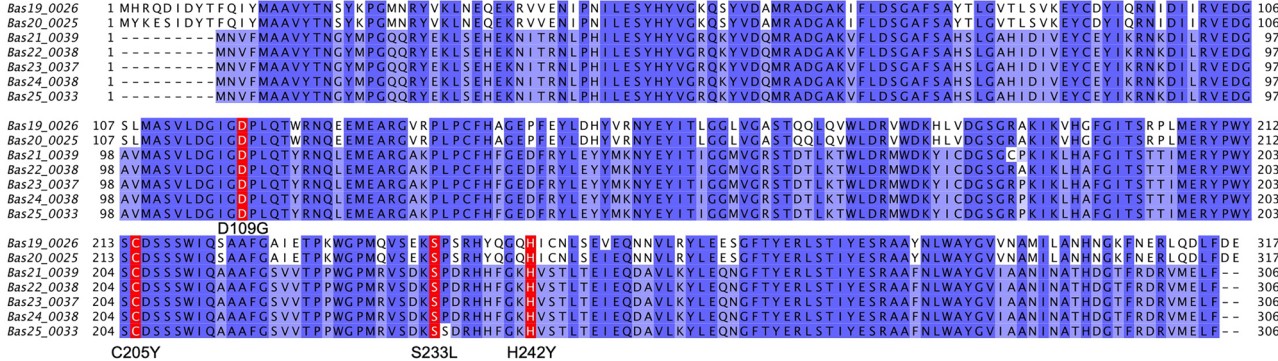

**Extended Data Fig. 9 | Bas24 escaper phages overcome GrwAB defence via mutations in the DNA modification pathway.** (**a**) Plaque assays evaluating the ability of 10-fold serial dilutions of Bas24 escaper phages to form plaques on *E. coli* MG1655Δ*araCBAD* in the absence (No system) and presence of GrwAB (+ GrwAB), as compared to the parental Bas24 stock and a control stock propagated in the absence of GrwAB. (**b**) SNP table summarising results of whole genome sequencing for each phage compared to the published Bas24 reference sequence. Light yellow (SNP), dark yellow (SNP targeting *dpdA* locus),

HP (hypothetical protein). (**c**) Multiple-sequence alignment showing conservation of *Queuovirinae* (Bas19-25) homologues of the 7-deazaguanosine inserting enzyme DpdA, with the residues found to be substituted in the Bas24 escaper phages highlighted in red. (**d**) Locations of the substituted residues shown on predicted structure of Bas24 DpdA, generated using AlphaFold3, compared to the experimentally determined crystal structure of *Salmonella enterica* serovar Montevideo DpdA (PDB 7UI4) showing the active site highlighted in magenta.

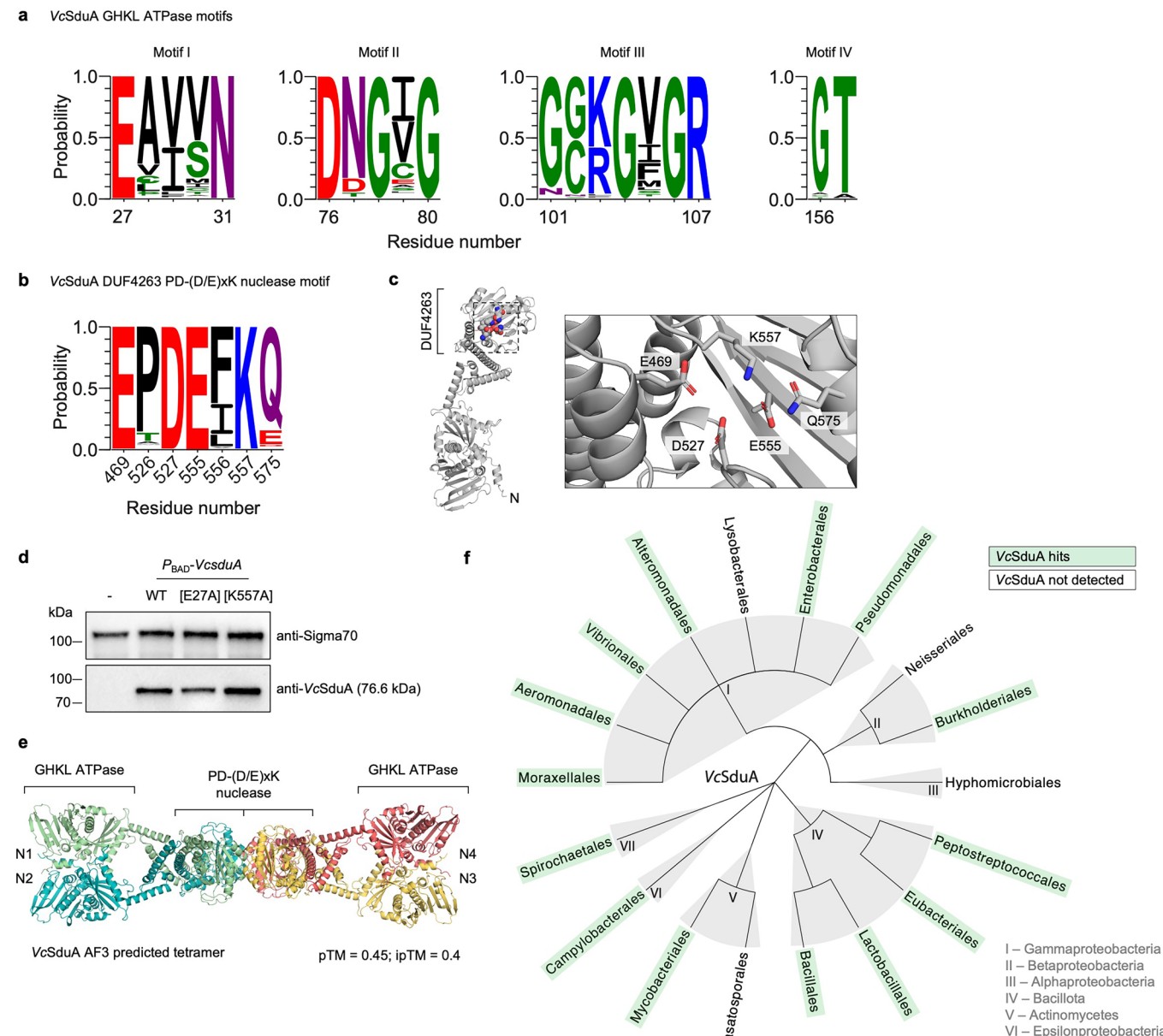

**Extended Data Fig. 10 | Identification of conserved motifs and distribution of *Vc*SduA.** (**a**-**b**) Sequence logos showing conservation of the identified GHKL ATPase motifs I-IV in the GHKL domain (**a**) and the identified PD-(D/E)xK nuclease motif in the DUF4263 domain (**b**) of the 543 *Vc*SduA hits detected using MacSyFinder v.2.1.1, as compared to the equivalent residue number in *V. cholerae* A1552 *Vc*SduA. Amino acids in logos are coloured according to chemical properties: polar (G, S, T, Y, C), green; neutral (Q, N), purple; basic (K, R, H), blue; acidic (D, E), red; and hydrophobic (A, V, L, I, P, W, F, M), black. (**c**) Location of the identified PD-(D/E)xK nuclease residues in the *Vc*SduA structural prediction. (**d**) Western blot showing the protein levels of *Vc*SduA,

expressed in *E. coli* MG1655Δ*araCBAD* from a chromosomally integrated transposon carrying the arabinose-inducible $P_{BAD}$- promoter, induced by the addition of 0.2% arabinose, as compared to derivatives encoding the indicated site-directed variants. Sigma70 was used as a loading control. Blots are representative of the results of three independent experiments. (**e**) AlphaFold3 predicted structure of potential *Vc*SduA tetramer. (**f**) Distribution of *Vc*SduA hits detected using MacSyFinder v.2.1.1. The tree shows the order-level phylogeny of genera in the RefSeq database with more than 500 genomes (see methods). For the full list of 543 *Vc*SduA hits see Supplementary Table 7.

                          David W. Adams

# Reporting Summary

## Statistics

For all statistical analyses, confirm that the following items are present in the figure legend, table legend, main text, or Methods section.

| n/a | Confirmed | |
|---|---|---|
| ☐ | ☒ | The exact sample size (*n*) for each experimental group/condition, given as a discrete number and unit of measurement |
| ☐ | ☒ | A statement on whether measurements were taken from distinct samples or whether the same sample was measured repeatedly |
| ☒ | ☐ | The statistical test(s) used AND whether they are one- or two-sided<br>*Only common tests should be described solely by name; describe more complex techniques in the Methods section.* |
| ☒ | ☐ | A description of all covariates tested |
| ☐ | ☒ | A description of any assumptions or corrections, such as tests of normality and adjustment for multiple comparisons |
| ☐ | ☒ | A full description of the statistical parameters including central tendency (e.g. means) or other basic estimates (e.g. regression coefficient) AND variation (e.g. standard deviation) or associated estimates of uncertainty (e.g. confidence intervals) |
| ☒ | ☐ | For null hypothesis testing, the test statistic (e.g. *F*, *t*, *r*) with confidence intervals, effect sizes, degrees of freedom and *P* value noted<br>*Give P values as exact values whenever suitable.* |
| ☒ | ☐ | For Bayesian analysis, information on the choice of priors and Markov chain Monte Carlo settings |
| ☒ | ☐ | For hierarchical and complex designs, identification of the appropriate level for tests and full reporting of outcomes |
| ☒ | ☐ | Estimates of effect sizes (e.g. Cohen's *d*, Pearson's *r*), indicating how they were calculated |

*Our web collection on statistics for biologists contains articles on many of the points above.*

## Software and code

Policy information about availability of computer code

| Data collection | Commercial software used for data collection was Zeiss Zen software (v.2.6 blue edition) |
|---|---|
| Data analysis | Commercial software used for data analysis was SnapGene v.4.3.11, Geneious Prime v.11.0.14.1, and LightCycler 96 software v.1.1.0.1320 (Roche). Open access software used for data analysis was ImageJ (version 2.1.0/1.53h; imagej.net/software/fiji), DefenseFinder v.1.2.4 Webserver (https://defensefinder.mdmlab.fr), PADLOC v.2.0.0 Webserver (https://padloc.otago.ac.nz/padloc/), MOTIF Search tool (genome.jp/tools/motif/), Jalview v.2.11.4.0, HHpred Webserver (https://toolkit.tuebingen.mpg.de/tools/hhpred; PDB_mmCIF70_8_Mar, default database), DALI Webserver (http://ekhidna2.biocenter.helsinki.fi/dali/), AlphaFold3 Webserver (https://alphafoldserver.com), Weblogo 3 Webserver (https://weblogo.threeplusone.com), clinker v.0.0.28, MacSyFinder v.2.1.1, MAFFT v.7.508, HMMER suite v.3.3.2, Unicycler v.0.5.0, snp-sites v.2.5.1, NCBI tool suite v.2.14.1, and flye v 2.9.3. |

For manuscripts utilizing custom algorithms or software that are central to the research but not yet described in published literature, software must be made available to editors and reviewers. We strongly encourage code deposition in a community repository (e.g. GitHub). See the Nature Portfolio guidelines for submitting code & software for further information.

## Data

Policy information about availability of data

All manuscripts must include a data availability statement. This statement should provide the following information, where applicable:

- Accession codes, unique identifiers, or web links for publicly available datasets
- A description of any restrictions on data availability
- For clinical datasets or third party data, please ensure that the statement adheres to our policy

> The genome assemblies of V. cholerae strain N19-2759 have been deposited in the NCBI GenBank database under accession number GCA_046097525.1. The raw reads are available from the Sequence Read Archive (SRA) under submission number SRX26909066. The publicly available datasets used in this work and the dates that they were accessed are listed in the methods. All other data are available in the main text or the supplementary materials. Source data are provided with this paper.

## Research involving human participants, their data, or biological material

Policy information about studies with human participants or human data. See also policy information about sex, gender (identity/presentation), and sexual orientation and race, ethnicity and racism.

| | |
|---|---|
| Reporting on sex and gender | n/a (study on bacteria) |
| Reporting on race, ethnicity, or other socially relevant groupings | n/a (study on bacteria) |
| Population characteristics | n/a (study on bacteria) |
| Recruitment | n/a (study on bacteria) |
| Ethics oversight | n/a (study on bacteria) |

Note that full information on the approval of the study protocol must also be provided in the manuscript.

# Field-specific reporting

Please select the one below that is the best fit for your research. If you are not sure, read the appropriate sections before making your selection.

☒ Life sciences    ☐ Behavioural & social sciences    ☐ Ecological, evolutionary & environmental sciences

For a reference copy of the document with all sections, see nature.com/documents/nr-reporting-summary-flat.pdf

# Life sciences study design

All studies must disclose on these points even when the disclosure is negative.

| | |
|---|---|
| Sample size | All experiments were performed independently three times, unless explicitly stated otherwise. No statistical methods were used to predetermine sample size. The use of multiple independent biological repeats to ensure reproducibility is standard practice in the field. |
| Data exclusions | No data were excluded from the analysis. |
| Replication | All experimental data are representative of the results of three independent biological repeats, except for the BASEL bacteriophage collection screening, which was repeated twice. All replication attempts were successful. |
| Randomization | Randomization is not standard practice in the field and was not employed. |
| Blinding | Blinding is not standard practice in the field and was not employed. |

# Reporting for specific materials, systems and methods

We require information from authors about some types of materials, experimental systems and methods used in many studies. Here, indicate whether each material, system or method listed is relevant to your study. If you are not sure if a list item applies to your research, read the appropriate section before selecting a response.

## Materials & experimental systems

| n/a | Involved in the study |
|---|---|
| ☐ | ☒ Antibodies |
| ☒ | ☐ Eukaryotic cell lines |
| ☒ | ☐ Palaeontology and archaeology |
| ☒ | ☐ Animals and other organisms |
| ☒ | ☐ Clinical data |
| ☒ | ☐ Dual use research of concern |
| ☒ | ☐ Plants |

## Methods

| n/a | Involved in the study |
|---|---|
| ☒ | ☐ ChIP-seq |
| ☒ | ☐ Flow cytometry |
| ☒ | ☐ MRI-based neuroimaging |

## Antibodies

| | |
|---|---|
| Antibodies used | Primary antibodies against WonA (2210455), WonB (2210453), and VcSduA (2310059) were custom-raised in rabbits against synthetic peptides (Eurogentec) and used at a dilution of 1:500. <br><br> Direct-Blot™ HRP anti-E. coli RNA Sigma 70 Antibody (663205, BioLegend, U.S.A) was used at a dilution of 1:20,000. <br><br> The Anti-Rabbit IgG (whole molecule)–Peroxidase secondary antibody (A9169, Sigma-Aldrich) was used at a dilution of 1:20,000. |
| Validation | The specificity of custom-raised antibodies was validated by the inclusion of negative control samples lacking the relevant proteins in the Western Blot analysis. <br><br> For the commercial secondary antibody the following validation statement was taken from the Sigma website: Specificity of the Peroxidase Conjugated Anti-Rabbit IgG antibodies was determined by immunoelectrophoresis (IEP) versus normal rabbit serum and rabbit IgG. <br><br> For the commercial Direct-Blot™ HRP anti-E. coli RNA Sigma 70 Antibody the following validation statement was taken from the BioLegend website: Each lot of this antibody is quality control tested by Western blotting. |

## Plants

| | |
|---|---|
| Seed stocks | *Report on the source of all seed stocks or other plant material used. If applicable, state the seed stock centre and catalogue number. If plant specimens were collected from the field, describe the collection location, date and sampling procedures.* |
| Novel plant genotypes | *Describe the methods by which all novel plant genotypes were produced. This includes those generated by transgenic approaches, gene editing, chemical/radiation-based mutagenesis and hybridization. For transgenic lines, describe the transformation method, the number of independent lines analyzed and the generation upon which experiments were performed. For gene-edited lines, describe the editor used, the endogenous sequence targeted for editing, the targeting guide RNA sequence (if applicable) and how the editor was applied.* |
| Authentication | *Describe any authentication procedures for each seed stock used or novel genotype generated. Describe any experiments used to assess the effect of a mutation and, where applicable, how potential secondary effects (e.g. second site T-DNA insertions, mosiacism, off-target gene editing) were examined.* |

