## [Peer Review File · Nature Microbiology]

West African South American pandemic *Vibrio cholerae* encodes multiple distinct phage defence systems

Corresponding Author: Professor Melanie Blokesch

Version 0:

Reviewer comments:

Reviewer #1

(Remarks to the Author)

In this manuscript, Adams et al characterize the *V. Cholera* WASA lineage, and identify multiple novel anti-phage immune systems. These include the WonAB system that confers immunity against the well-characterized ICP1, an R/M-like system, and a Shedu system. This work is timely and important, building upon previous discoveries of *V. cholerae* immune systems by the Blokesch group and others. I strongly recommend publication of this work in Nature Microbiology if the authors are able to address my minor concerns.

1. Structural predictions.

I appreciate the efforts the authors have gone to to provide AlphaFold models for the novel systems, and that they have included critical validation metrics (pTM and iPTM). However, this could still be improved. The authors should add a supplementary figure where the structural predictions (at least the ones for the main figures, but the WonA2B(1 or 2) complex would be useful) are shown colored by per-residue confidence score. This is particularly important for the reader to judge interfaces within oligomeric assemblies (especially the GHKL domain interface for VcSduA which I personally believe is valid when compared to other GHKL domain dimers (e.g. Hsp90 in PDB 5FWK or many others) but appears to be mediated by a fairly limited interface as judged by eye in Fig 4F).

2. Mechanism of WonAB

Since WonAB appears to be a substantial aspect of the manuscript, could the authors speculate more directly about the mechanism of anti-phage defense by this system? The authors demonstrate that WonAB shuts down cell growth through translation inhibition, but is there any homology between WonAB and other translation inhibitors? Do you think WonAB is cleaving mRNA non-specifically, tRNAs or is it targeting the ribosome? The latter two scenarios are raised as possibilities, but perhaps alignment of WonAB models with other mRNA/tRNA/rRNA nuclease structures may reveal further mechanistic insights into this system. It would also be useful to include a predicted structure for the WonAB complex in the main figure, since they are likely to interact (based on the ipTM score).

The authors may also include a schematic for WonAB mechanism in Fig 3 since this exciting system seems to have a novel anti-phage mechanism within the arsenal of post-Doron 2018 bacterial immune systems, and it would be a pity for WonAB to get overlooked in the vast amount of literature that continues to be published on this topic.

3. VcSduA

The discovery of a novel Shedu variant is indeed interesting. However, previous Shedu structure have shown a tetrameric state - have the authors attempted folding VcShuA as a tetramer? The authors speculate that GHKL is a sensor, but it is unclear what this is a sensor of? In other systems such as Hsp90 or DNA gyrase, GHKL functions as a clasp that opens or closes depending on bound nucleotide. I appreciate that it may be premature to speculate on the activator of VcSduA, but I would refrain from referring to this domain as a sensor.

4. GrwAB

Did the authors predict structures of GrwAB complexes with alternative stoichiometries? The authors conclude that GrwA and GrwB function together, and should include a supplementary figure to show these predictions, even if they are not predicted with high confidence.

Jack Bravo

Reviewer #2

(Remarks to the Author)

Here, Adams et. al. presents a very nice manuscript describing novel phage defense systems in the West African Sout American (WASA) pandemic *Vibrio cholerae* (Vc) strains. The most exciting finding of the authors is that the WASA lysogenic prophage characteristic of these strains encodes a novel phage defense systems that the authors labeled WonAB. They show that WonAB consists of a putative nuclease and ATPase and present some data that suggests this system inhibits translation. However, the mechanism of this system is still not well worked out as the authors do not perform any biochemistry to analyze the activity of these proteins nor define the targets. WonAB does protect Vc against ICP-1 phage, which is an exciting finding as this phage seems to be the major phage predator of Vc during infection. The authors also describe two other new phage defense systems encoded on the VSP-II island and identify *E. coli* phage that are sensitive to these systems. But these systems are similar to others that have already been described, and the authors do not really characterize any mechanism nor role of these systems against cholera phage. Thus, the experiments are quite well done and clearly explained, and the findings are an important contribution to the field, but my enthusiasm is somewhat dampened by a lack of understanding of WonAB function and less novelty with the two VSP-II phage defense systems. Specific comments:

1. Line 74-the wordings.” broadly conserved genome encoding proteins...” is unclear. Please clarify.
2. Lines 132-133-I am curious if these delayed lysis cells release viable phage. The next sentence implies that the culture rebounded due to the evolution of ICP-1 resistance, which suggests that perhaps they do and ICP-1 infection is not completely inhibited.
3. Line 137- Have the authors tried to isolate escape mutants in *E. coli* using the BASEL phage?
4. A straightforward experiment to test if WonAB inhibits translation more directly is to add purified WonAB to an in vitro translation assay. If it targets tRNAs, it should block translation.
5. The discovery of a new Type IV restriction system in the VSP-2 island is a novel contribution. But, considering it is similar (although not identical as the authors describe) conceptually and in many ways structural (again not identical but similar), I would not consider this a high impact finding. Moreover, the authors do not provide any evidence for the role of this system in the evolution of Vc or the WASA lines, which, considering this is a major thrust of the manuscript, dampens my enthusiasm for the impact of the finding.
6. Do any of the known Vc phages encode a DpdA ortholog that might be tarageted by the Type IV system?
7. Similar to my comment #5, the discovery of the Vc SduA system is interesting, but I would not regard it has high impact as these systems are known and there is not really any new knowledge beyond the discovery of this system in Vc and the authors do not connect the presence of this system to protection to any Vc phage.
8. Given that WonAB is on the WASA prophage and it protects against ICP-1, why wasn't it maintained in other Vc pandemic spreading events? Does it protect against current ICP-1 phage or did ICP-1 evolve resistance or countermeasures?

Reviewer #3

(Remarks to the Author)

There is a surge of discoveries of defense systems used to protect against bacteriophage infection, with many encoded on prophages within bacterial genomes, as reviewed recently (Murtazaliev et al 2024 <https://doi.org/10.1016/j.tim.2024.05.005>; Patel & Maxwell, 2023 (Ref 30) <https://doi.org/10.1016/j.mib.2023.102321>; Millman et al , 2022. <https://doi.org/10.1016/j.chom.2022.09.017>; Rocha & Bikard, 2022. <https://doi.org/10.1371/journal.pbio.3001514>). The manuscript by Adams et al describes the characterization of three bacteriophage defense systems in a lineage of the bacterium *Vibrio cholerae*, which can cause cholera pandemics. The authors challenged several Peruvian *V. cholerae* strains with three lytic phages ICP1-3, and discovered they could not be infected by standard plate plaque assays. The Peruvian strain behave similarly to other strains with known phage defence mechanisms, and have encode two previously described genetic loci – a VSP-II locus and a prophage called WASA-1. They discovered the WASA-1 prophage prevents ICP1 phage infection and encodes two genes WonA (a nuclease) and WonB that may function together (WonAB) as a novel defence mechanism, with homologs found in a small number of other bacteria. The authors propose WonAB promotes abortive infection: altruistic bacterial cell death that leads to fewer viruses produced. The manuscript also includes more brief description of two other loci found in the VSP-II locus: of a restriction modification system (GrwAB) and a Shedu defence system (VcSduA). Both are also found in ~1% of other bacteria. The authors are well qualified to conduct these experiments. The technical aspects of the manuscript are conducted carefully with genetic manipulation, bioinformatics, microscopy and video documentation. The impact of the manuscript is perhaps diminished by the recent dearth of manuscripts in this area. Also, there are many important recent references lacking that can bolster the manuscript and add an ecological and evolutionary perspective as described below.

Major comments

Line 127. How do the results presented at high MOI distinguish between abortive infection and killing by the phage?

Fig 1A and 1C. The plaques do not dilute out serially on the Peru strains. ---The authors might consider assessing the phage population over time in liquid cultures as described in <https://doi.org/10.1101/2024.07.10.602838>.

Fig 2A. What was the rationale for running the time-course for only 5 hours (300 minutes)? Might you expect recovery if this was measured for longer (ex: 24h)?

Fig 3. Why was the experiments terminated at 3 hours when the hypothesis is that WonAB inhibits translation and does not induce lysis. Was the experiment extended for longer time periods?

Minor comments:

Line 18 replace “predatory” with “lytic”

Line 26 please provide citations

Line replace “predation” with “lysis”

Line 32 – Why is there an urgent need? Is evolved resistance to phage observed? Common? If yes, what kind of resistance? Surface receptor-based?

Line 42 – “...with the emergence ...” is confusing and should be reworded. In particular “resistant ICP1” is awkward since phage resistance is commonly ascribed to the bacterial host not the virus. We suggest rewording to clarify what is being described is the emergence of ICP1 capable of infecting a phage resistant host.

Line 45 –A review of host-phage interactions (e.g. <https://doi.org/10.1038/s41564-024-01832-5>) would useful for a reader here in the introduction. Prior work suggests defenses play a minor role compared to surface modifications. Have Vibrio surface modifications been reported that promote phage resistance?

Line 59 – Did the authors looks for any evidence of phenotypic changes that might result from modification of cell surface receptors?

Line 137, 223. Perhaps for an audience less familiar with ICP1, the author could remind readers that ICP1 is a phage. For example, consider rewording to replace “ICP1 escape mutant”, “escapers” and “escaper mutants” with “ICP1 escape phage”. This would remind the audience that these mutations are in the phage not the bacterial genome.

Line 105 and 118-120. WonAB is found in 0.43% of bacterial genomes. Similarly, line 233 describes GrwAB as “widespread”? At 1%? In line 260 VcSduA homologs are noted as “widespread”. At ~1.2%? A frank recognition that the systems described are relatively rare is warranted.

Lines 133-136. The authors should unpack this result more. Is the candidate defence system simply delaying lysis and allowing for emergence of spontaneous O1 antigen receptor mutants? What do the authors mean by in vitro here? Outside the human host? In the laboratory? Are O1-antigen mutants observed in *V. cholerae* derived from non-pandemic sources?

Line 192 This section beginning here would benefit from a brief reminder of the initial hypothesis to explore VSP-II.

Line 243. This section on VcSduA would benefit from an introduction.

Lines 265-266. Some context would be helpful here. This section reads more like a list - we found two more.

Line 276. It is fascinating that many anti-phage systems are encoded on prophages. The impact of the manuscript could be improved by testing (comments for lines 141-142) and discussing the implications of WonAB being on the WASA-1 prophage itself. The lifecycle of the well studied *V. cholerae* CTX phage has profound effects on the dynamics of the *Vibrio* population as well as the phage itself. A more in-depth exploration of the consequences of this arrangement would be a welcome addition.

Line 287. The authors should draw on a wealth of citation to discuss how prophages and the genetic material they carry can help their bacterial hosts “succeed”.

Line 281. Citations describing “superinfection exclusion” are needed here as well as use of this term in the manuscript. It is well described in the literature (ex: <https://doi.org/10.1371/journal.pcbi.1010125>) but not referred to anywhere in the manuscript.

Lines 297-300. Again, the Discussion can be improved by mention of ecological and evolutionary theories regarding mobile genetic elements. Several recent reviews include: <https://doi.org/10.1016/j.mib.2024.102436>; <https://doi.org/10.1038/s41467-024-46489-0>;

Lines 141-142. Is WonAB activated in response to other stressor beyond phage? It would be interesting to know whether ICP1 induces the WASA-1 WonAB-containing prophage.

Lines 87-88 – Do prophages carry antiphage defence system to increase their host's fitness? The results suggest “superinfection exclusion” (<https://doi.org/10.1371/journal.pcbi.1010125>), which selfishly benefits the prophage by preventing infection by other phage. The manuscript's impact can be improved by recognition and integration of current work exploring the evolutionary consequences of phage co-infection on community composition. Indeed there are two recent papers on phage dynamics in *Vibrio* populations that need to be included here: Hussain, et al 2021 (<https://doi.org/10.1126/science.abb1083>, Piel et al 2022 (<https://doi.org/10.1038/s41564-022-01157-1>).

Brian K Hammer, PhD
Ellinor O Alseth, PhD

Decision Letter:

21st November 2024

Dear Melanie,

I hope you've been well. Thank you, as always, for your patience while your manuscript "Diverse phage defence systems define West African South American pandemic *Vibrio cholerae*" was under peer-review at Nature Microbiology. It has now been seen by 3 referees, whose expertise and comments you will find at the end of this email. I'm pleased to be able to share some encouraging news about your work. The reviewers find your manuscript of considerable potential interest, though they have raised a few concerns that will need to be addressed before we can consider publication of the work in Nature Microbiology.

There are no major technical issues, and the majority of the points raised are very straightforward and should be relatively simple to address. One important thing that did come up is the request for more mechanistic characterization of WonAB. The reviewers were very excited about this, and they felt that more insights here would substantially strengthen the manuscript. From our end, we certainly agree--if you could add more here, that would be excellent. However, I do want to assure you that a fully flushed out biochemical characterization is not something that we'd hold you to (and we will explain our editorial bar in this regard to the reviewers when we send back the revised manuscript, just so we're all on the same page).

Should further experimental data allow you to address these criticisms, we would be happy to look at a revised manuscript.

Please include a data availability statement as a separate section after Methods but before references, under the heading "Data Availability". This section should inform readers about the availability of the data used to support the conclusions of your study. This information includes accession codes to public repositories (data banks for protein, DNA or RNA sequences, microarray, proteomics data etc...), references to source data published alongside the paper, unique identifiers such as URLs to data repository entries, or data set DOIs, and any other statement about data availability. At a minimum, you should include the following statement: "The data that support the findings of this study are available from the corresponding author upon request", mentioning any restrictions on availability. If DOIs are provided, we also strongly encourage including these in the Reference list (authors, title, publisher (repository name), identifier, year). For more guidance on how to write this section please see: <http://www.nature.com/authors/policies/data/data-availability-statements-data-citations.pdf>

* If you have not done so already we suggest that you begin to revise your manuscript so that it conforms to our Article format instructions at <http://www.nature.com/nmicrobiol/info/final-submission>. Refer also to any guidelines provided in this letter.

When submitting the revised version of your manuscript, please pay close attention to our [href="https://www.nature.com/nature-portfolio/editorial-policies/image-integrity">Digital Image Integrity Guidelines](https://www.nature.com/nature-portfolio/editorial-policies/image-integrity) and to the following points below:

Link Redacted

Note: This url links to your confidential homepage and associated information about manuscripts you may have submitted or be reviewing for us. If you wish to forward this e-mail to co-authors, please delete this link to your homepage first.

Nature Microbiology is committed to improving transparency in authorship. As part of our efforts in this direction, we are now

requesting that all authors identified as 'corresponding author' on published papers create and link their Open Researcher and Contributor Identifier (ORCID) with their account on the Manuscript Tracking System (MTS), prior to acceptance. This applies to primary research papers only. ORCID helps the scientific community achieve unambiguous attribution of all scholarly contributions. You can create and link your ORCID from the home page of the MTS by clicking on 'Modify my Springer Nature account'. For more information please visit www.springernature.com/orcid.

If you wish to submit a suitably revised manuscript we would hope to receive it within 6 months. If you cannot send it within this time, please let us know. We will be happy to consider your revision, even if a similar study has been accepted for publication at Nature Microbiology or published elsewhere (up to a maximum of 6 months).

Yours sincerely,

Reviewer Expertise:

Referee #1: phage defense

Referee #2: phage defense, Vbriio

Referee #3: Vibrio cholerae

Reviewer Comments:

Reviewer #1 (Remarks to the Author):

In this manuscript, Adams et al characterize the V. Cholera WASA lineage, and identify multiple novel anti-phage immune systems. These include the WonAB system that confers immunity against the well-characterized ICP1, an R/M-like system, and a Shedu system. This work is timely and important, building upon previous discoveries of V. cholerae immune systems by the Blokesch group and others. I strongly recommend publication of this work in Nature Microbiology if the authors are able to address my minor concerns.

1. Structural predictions.

I appreciate the efforts the authors have gone to to provide AlphaFold models for the novel systems, and that they have included critical validation metrics (pTM and iPTM). However, this could still be improved. The authors should add a supplementary figure where the structural predictions (at least the ones for the main figures, but the WonA2B(1 or 2) complex would be useful) are shown colored by per-residue confidence score. This is particularly important for the reader to judge interfaces within oligomeric assemblies (especially the GHKL domain interface for VcSduA which I personally believe is valid when compared to other GHKL domain dimers (e.g. Hsp90 in PDB 5FWK or many others) but appears to be mediated by a fairly limited interface as judged by eye in Fig 4F).

2. Mechanism of WonAB

Since WonAB appears to be a substantial aspect of the manuscript, could the authors speculate more directly about the mechanism of anti-phage defense by this system? The authors demonstrate that WonAB shuts down cell growth through translation inhibition, but is there any homology between WonAB and other translation inhibitors? Do you think WonAB is cleaving mRNA non-specifically, tRNAs or is it targeting the ribosome? The latter two scenarios are raised as possibilities, but perhaps alignment of WonAB models with other mRNA/tRNA/rRNA nuclease structures may reveal further mechanistic insights into this system. It would also be useful to include a predicted structure for the WonAB complex in the main figure, since they are likely to interact (based on the ipTM score).

The authors may also include a schematic for WonAB mechanism in Fig 3 since this exciting system seems to have a novel anti-phage mechanism within the arsenal of post-Doron 2018 bacterial immune systems, and it would be a pity for WonAB to get overlooked in the vast amount of literature that continues to be published on this topic.

3. VcSduA

The discovery of a novel Shedu variant is indeed interesting. However, previous Shedu structure have shown a tetrameric state - have the authors attempted folding VcShuA as a tetramer? The authors speculate that GHKL is a sensor, but it is unclear what this is a sensor of? In other systems such as Hsp90 or DNA gyrase, GHKL functions as a clasp that opens or closes depending on bound nucleotide. I appreciate that it may be premature to speculate on the activator of VcSduA, but I would refrain from referring to this domain as a sensor.

4. GrwAB

Did the authors predict structures of GrwAB complexes with alternative stoichiometries? The authors conclude that GrwA and GrwB function together, and should include a supplementary figure to show these predictions, even if they are not predicted with high confidence.

Jack Bravo

Reviewer #2 (Remarks to the Author):

Here, Adams et. al. presents a very nice manuscript describing novel phage defense systems in the West African Sout American (WASA) pandemic *Vibrio cholerae* (Vc) strains. The most exciting finding of the authors is that the WASA lysogenic prophage characteristic of these strains encodes a novel phage defense systems that the authors labeled WonAB. They show that WonAB consists of a putative nuclease and ATPase and present some data that suggests this system inhibits translation. However, the mechanism of this system is still not well worked out as the authors do not perform any biochemistry to analyze the activity of these proteins nor define the targets. WonAB does protect Vc against ICP-1 phage, which is an exciting finding as this phage seems to be the major phage predator of Vc during infection. The authors also describe two other new phage defense systems encoded on the VSP-II island and identify *E. coli* phage that are sensitive to these systems. But these systems are similar to others that have already been described, and the authors do not really characterize any mechanism nor role of these systems against cholera phage. Thus, the experiments are quite well done and clearly explained, and the findings are an important contribution to the field, but my enthusiasm is somewhat dampened by a lack of understanding of WonAB function and less novelty with the two VSP-II phage defense systems. Specific comments:

1. Line 74-the wordings: ". broadly conserved genome encoding proteins..." is unclear. Please clarify.
2. Lines 132-133-I am curious if these delayed lysis cells release viable phage. The next sentence implies that the culture rebounded due to the evolution of ICP-1 resistance, which suggests that perhaps they do and ICP-1 infection is not completely inhibited.
3. Line 137- Have the authors tried to isolate escape mutants in *E. coli* using the BASEL phage?
4. A straightforward experiment to test if WonAB inhibits translation more directly is to add purified WonAB to an in vitro translation assay. If it targets tRNAs, it should block translation.
5. The discovery of a new Type IV restriction system in the VSP-2 island is a novel contribution. But, considering it is similar (although not identical as the authors describe) conceptually and in many ways structural (again not identical but similar), I would not consider this a high impact finding. Moreover, the authors do not provide any evidence for the role of this system in the evolution of Vc or the WASA lines, which, considering this is a major thrust of the manuscript, dampens my enthusiasm for the impact of the finding.
6. Do any of the known Vc phages encode a DpdA ortholog that might be tarageted by the Type IV system?
7. Similar to my comment #5, the discovery of the Vc SduA system is interesting, but I would not regard it has high impact as these systems are known and there is not really any new knowledge beyond the discovery of this system in Vc and the authors do not connect the presence of this system to protection to any Vc phage.
8. Given that WonAB is on the WASA prophage and it protects against ICP-1, why wasn't it maintained in other Vc pandemic spreading events? Does it protect against current ICP-1 phage or did ICP-1 evolve resistance or countermeasures?

Reviewer #3 (Remarks to the Author):

There is a surge of discoveries of defense systems used to protect against bacteriophage infection, with many encoded on prophages within bacterial genomes, as reviewed recently (Murtazaliev et al 2024 <https://doi.org/10.1016/j.tim.2024.05.005>; Patel & Maxwell, 2023 (Ref 30) <https://doi.org/10.1016/j.mib.2023.102321>; Millman et al , 2022. <https://doi.org/10.1016/j.chom.2022.09.017>; Rocha & Bikard, 2022. <https://doi.org/10.1371/journal.pbio.3001514>). The manuscript by Adams et al describes the characterization of three bacteriophage defense systems in a lineage of the bacterium *Vibrio cholerae*, which can cause cholera pandemics. The authors challenged several Peruvian *V. cholerae* strains with three lytic phages ICP1-3, and discovered they could not be infected by standard plate plaque assays. The Peruvian strain behave similarly to other strains with known phage defence mechanisms, and have encode two previously described genetic loci – a VSP-II locus and a prophage called WASA-1. They discovered the WASA-1 prophage prevents ICP1 phage infection and encodes two genes WonA (a nuclease) and WonB that may function together (WonAB) as a novel defence mechanism, with homologs found in a small number of other bacteria. The authors propose WonAB promotes abortive infection: altruistic bacterial cell death that leads to fewer viruses produced. The manuscript also includes more brief description of two other loci found in the VSP-II locus: of a restriction modification system (GrwAB) and a Shedu defence system (VcSduA). Both are also found in ~1% of other bacteria. The authors are well qualified to conduct these experiments. The technical aspects of the manuscript are conducted carefully with genetic manipulation, bioinformatics, microscopy and video documentation. The impact of the manuscript is perhaps diminished by the recent dearth of manuscripts in this area. Also, there are many important recent references lacking that can bolster the manuscript and add an ecological and evolutionary perspective as described below.

Major comments

Line 127. How do the results presented at high MOI distinguish between abortive infection and killing by the phage?

Fig 1A and 1C. The plaques do not dilute out serially on the Peru strains. The authors might consider assessing the phage population over time in liquid cultures as described in <https://doi.org/10.1101/2024.07.10.602838>.

Fig 2A. What was the rationale for running the time-course for only 5 hours (300 minutes)? Might you expect recovery if this was measured for longer (ex: 24h)?

Fig 3. Why was the experiments terminated at 3 hours when the hypothesis is that WonAB inhibits translation and does not induce lysis. Was the experiment extended for longer time periods?

Minor comments:

Line 18 replace “predatory” with “lytic”

Line 26 please provide citations

Line replace “predation” with “lysis”

Line 32 – Why is there an urgent need? Is evolved resistance to phage observed? Common? If yes, what kind of resistance? Surface receptor-based?

Line 42 – “...with the emergence ...” is confusing and should be reworded. In particular “resistant ICP1” is awkward since phage resistance is commonly ascribed to the bacterial host not the virus. We suggest rewording to clarify what is being described is the emergence of ICP1 capable of infecting a phage resistant host.

Line 45 –A review of host-phage interactions (e.g. <https://doi.org/10.1038/s41564-024-01832-5>) would be useful for a reader here in the introduction. Prior work suggests defenses play a minor role compared to surface modifications. Have Vibrio surface modifications been reported that promote phage resistance?

Line 59 – Did the authors look for any evidence of phenotypic changes that might result from modification of cell surface receptors?

Line 137, 223. Perhaps for an audience less familiar with ICP1, the author could remind readers that ICP1 is a phage. For example, consider rewording to replace “ICP1 escape mutant”, “escapers” and “escaper mutants” with “ICP1 escape phage”. This would remind the audience that these mutations are in the phage not the bacterial genome.

Line 105 and 118-120. WonAB is found in 0.43% of bacterial genomes. Similarly, line 233 describes GrwAB as “widespread”? At 1%? In line 260 VcSduA homologs are noted as “widespread”. At ~1.2%? A frank recognition that the systems described are relatively rare is warranted.

Lines 133-136. The authors should unpack this result more. Is the candidate defence system simply delaying lysis and allowing for emergence of spontaneous O1 antigen receptor mutants? What do the authors mean by in vitro here? Outside the human host? In the laboratory? Are O1-antigen mutants observed in *V. cholerae* derived from non-pandemic sources?

Line 192 This section beginning here would benefit from a brief reminder of the initial hypothesis to explore VSP-II.

Line 243. This section on VcSduA would benefit from an introduction.

Lines 265-266. Some context would be helpful here. This section reads more like a list - we found two more.

Line 276. It is fascinating that many anti-phage systems are encoded on prophages. The impact of the manuscript could be improved by testing (comments for lines 141-142) and discussing the implications of WonAB being on the WASA-1 prophage itself. The lifecycle of the well studied *V. cholerae* CTX phage has profound effects on the dynamics of the *Vibrio* population as well as the phage itself. A more in-depth exploration of the consequences of this arrangement would be a welcome addition.

Line 287. The authors should draw on a wealth of citation to discuss how prophages and the genetic material they carry can help their bacterial hosts “succeed”.

Line 281. Citations describing “superinfection exclusion” are needed here as well as use of this term in the manuscript. It is well described in the literature (ex: <https://doi.org/10.1371/journal.pcbi.1010125>) but not referred to anywhere in the manuscript.

Lines 297-300. Again, the Discussion can be improved by mention of ecological and evolutionary theories regarding mobile genetic elements. Several recent reviews include: <https://doi.org/10.1016/j.mib.2024.102436>; <https://doi.org/10.1038/s41467-024-46489-0>;

Lines 141-142. Is WonAB activated in response to other stressor beyond phage? It would be interesting to know whether ICP1 induces the WASA-1 WonAB-containing prophage.

Lines 87-88 – Do prophages carry antiphage defence system to increase their host’s fitness? The results suggest “superinfection exclusion” (<https://doi.org/10.1371/journal.pcbi.1010125>), which selfishly benefits the prophage by preventing infection by other phage. The manuscript’s impact can be improved by recognition and integration of current work exploring the evolutionary consequences of phage co-infection on community composition. Indeed there are two recent papers on phage dynamics in *Vibrio* populations that need to be included here: Hussain, et al 2021 (<https://doi.org/10.1126/science.abb1083>), Piel et al 2022 (<https://doi.org/10.1038/s41564-022-01157-1>).

Brian K Hammer, PhD
Ellinor O Alseth, PhD

Version 1:

Reviewer comments:

Reviewer #1

(Remarks to the Author)

I am happy with the changes to the manuscript, and I fully support the publication of this fascinating work. The (predicted) 2:1 stoichiometry of the WonAB complex is particularly fascinating, and I look forward to future mechanistic work establishing how WonAB and the other WASA defense systems function.

I will add that I agree with the authors regarding in-depth mechanistic studies (response to reviewer #2) - I think it is completely appropriate to have separate discovery and mechanistic manuscripts, especially where multiple previously-uncharacterized systems are discovered (i.e. here).

Jack Bravo

Reviewer #3

(Remarks to the Author)

Firstly, we thank the authors for providing responses for each concern raised, which is much appreciated. Most of the replies, however, referred to requirements to compress the text, and due to this brevity, some of the novelty of the findings are still not articulated fully. Several of our comments and concerns were echoed by Reviewer #2 as well. As stated, we feel the technical aspects of the manuscript are sound, and the authors' expertise on the topic is clearly reflected in their writing. The authors make some interesting claims regarding potential contribution to cholera pandemic successes, but the brevity of the manuscript diminish the support for such statements, as well as the impacts and importance of the study for a broad audience.

Brian Hammer and Ellinor Alseth

Decision Letter:

Our ref: NMICROBIOL-24103108A

24th February 2025

Dear Melanie,

Thank you for submitting your revised manuscript "Diverse phage defence systems define West African South American pandemic *Vibrio cholerae*" (NMICROBIOL-24103108A). It has now been seen by the original referees and their comments are below. The reviewers find that the paper has improved in revision, and therefore we'll be happy in principle to publish it in Nature Microbiology, pending minor revisions to satisfy the referees' final requests and to comply with our editorial and formatting guidelines.

Looking forward to raising a toast to this paper when we're in Paris for the conference!

Thank you again for your interest in Nature Microbiology Please do not hesitate to contact me if you have any questions.

Sincerely,

Reviewer #1 (Remarks to the Author):

I am happy with the changes to the manuscript, and I fully support the publication of this fascinating work. The (predicted) 2:1 stoichiometry of the WonAB complex is particularly fascinating, and I look forward to future mechanistic work establishing how WonAB and the other WASA defense systems function.

I will add that I agree with the authors regarding in-depth mechanistic studies (response to reviewer #2) - I think it is completely appropriate to have separate discovery and mechanistic manuscripts, especially where multiple previously-uncharacterized

systems are discovered (i.e. here).

Jack Bravo

Reviewer #3 (Remarks to the Author):

Firstly, we thank the authors for providing responses for each concern raised, which is much appreciated. Most of the replies, however, referred to requirements to compress the text, and due to this brevity, some of the novelty of the findings are still not articulated fully. Several of our comments and concerns were echoed by Reviewer #2 as well. As stated, we feel the technical aspects of the manuscript are sound, and the authors' expertise on the topic is clearly reflected in their writing. The authors make some interesting claims regarding potential contribution to cholera pandemic successes, but the brevity of the manuscript diminish the support for such statements, as well as the impacts and importance of the study for a broad audience.

Brian Hammer and Ellinor Alseth

Version 2:

Decision Letter:

3rd April 2025

Dear Melanie,

I am pleased to accept your Article "West African South American pandemic *Vibrio cholerae* encodes multiple distinct phage defence systems" for publication in Nature Microbiology. Thank you for having chosen to submit your work to us and many congratulations. I'm thrilled we'll get to raise a glass to this one in Paris next week!

Authors may need to take specific actions to achieve [compliance](https://www.springernature.com/gp/open-research/funding/policy-compliance-faqs) with funder and institutional open access mandates. If your research is supported by a funder that requires immediate open access (e.g. according to [Plan S principles](https://www.springernature.com/gp/open-research/plan-s-compliance)) then you should select the gold OA route, and we will direct you to the compliant route where possible. For authors selecting the subscription publication route, the journal's standard licensing terms will need to be accepted, including [self-archiving policies](https://www.nature.com/nature-portfolio/editorial-policies/self-archiving-and-license-to-publish). Those licensing terms will supersede any other terms that the author or any third party may assert apply to any version of the manuscript.

With kind regards,

P.S. Click on the following link if you would like to recommend Nature Microbiology to your librarian
<http://www.nature.com/subscriptions/recommend.html#forms>

** Visit the Springer Nature Editorial and Publishing website at http://editorial-jobs.springernature.com?utm_source=ejP_NMicro_email&utm_medium=ejP_NMicro_email&utm_campaign=ejp_NMicro for more information about our career opportunities. If you have any questions please click [here](mailto:editorial.publishing.jobs@springernature.com).

Open Access This Peer Review File is licensed under a Creative Commons Attribution 4.0 International License, which permits use, sharing, adaptation, distribution and reproduction in any medium or format, as long as you give appropriate credit to the original author(s) and the source, provide a link to the Creative Commons license, and indicate if changes were made. In cases where reviewers are anonymous, credit should be given to 'Anonymous Referee' and the source. The images or other third party material in this Peer Review File are included in the article's Creative Commons license, unless indicated otherwise in a credit line to the material. If material is not included in the article's Creative Commons license and your intended use is not permitted by statutory regulation or exceeds the permitted use, you will need to obtain permission directly from the copyright holder.

Response to referees

First and foremost, we thank the editor and the three reviewers for their positive appraisals of our work and their helpful comments. We are confident that we have addressed all points of critique, as outlined below. The reviewers' comments are in blue, and our responses are in black.

Reviewer 1 (Remarks to the Author):

#1) In this manuscript, Adams et al characterize the *V. Cholera* WASA lineage, and identify multiple novel anti-phage immune systems. These include the WonAB system that confers immunity against the well-characterized ICP1, an R/M-like system, and a Shedu system. This work is timely and important, building upon previous discoveries of *V. cholerae* immune systems by the Blokesch group and others. I strongly recommend publication of this work in *Nature Microbiology* if the authors are able to address my minor concerns.

Authors' reply #1: We sincerely thank the reviewer for their accurate summary of our work and their kind and encouraging feedback.

#2) 1. Structural predictions.

I appreciate the efforts the authors have gone to to provide AlphaFold models for the novel systems, and that they have included critical validation metrics (pTM and iPTM). However, this could still be improved. The authors should add a supplementary figure where the structural predictions (at least the ones for the main figures, but the WonA2B(1 or 2) complex would be useful) are shown colored by per-residue confidence score. This is particularly important for the reader to judge interfaces within oligomeric assemblies (especially the GHKL domain interface for VcSduA which I personally believe is valid when compared to other GHKL domain dimers (e.g. Hsp90 in PDB 5FWK or many others) but appears to be mediated by a fairly limited interface as judged by eye in Fig 4F).

Authors' reply #2: We thank the reviewer for these valuable suggestions. As suggested, we have now included the full confidence metrics for all the AlphaFold models in the newly added Supplementary Figure 1.

#3) 2. Mechanism of WonAB

Since WonAB appears to be a substantial aspect of the manuscript, could the authors speculate more directly about the mechanism of anti-phage defense by this system? The authors demonstrate that WonAB shuts down cell growth through translation inhibition, but is there any homology between WonAB and other translation inhibitors? Do you think WonAB is cleaving mRNA non-specifically, tRNAs or is it targeting the ribosome? The latter two scenarios are raised as possibilities, but perhaps alignment of WonAB models with other mRNA/tRNA/rRNA nuclease structures may reveal further mechanistic insights into this system. It would also be

useful to include a predicted structure for the WonAB complex in the main figure, since they are likely to interact (based on the ipTM score).

Authors' reply #3: As noted by the reviewer, our results support a model whereby upon activation WonAB aborts phage infection by inhibiting translation, and thus prevents the progression of the ICP1 lifecycle. Furthermore, given the lack of apparent DNA damage, we speculate in the text that this could be achieved by targeting a rRNA or tRNA. Indeed, several defence systems utilise an ABC-ATPase domain like that found in WonA to regulate tRNA targeting nucleases, such as TOPRIM in PARIS (Burman *et al.*, 2024 PMID: 39111359 & Deep *et al.*, 2024 PMID: 39112702) the HEPN in PrrC, RloC and others (Krishnan *et al.*, 2020 PMID: 32894288) and the HNH nuclease domain of a Retron-associated Septu system (Azam *et al.*, 2024 PMID: 39528469). However, to our knowledge no systems containing the PD-D/ExK domain of WonB have so far been implicated in translation inhibition, thus precluding a direct alignment. Finally, as suggested, we have moved the predicted structure of the WonAB complex to the main figure (Fig. 1G).

#4) The authors may also include a schematic for WonAB mechanism in Fig 3 since this exciting system seems to have a novel anti-phage mechanism within the arsenal of post-Doron 2018 bacterial immune systems, and it would be a pity for WonAB to get overlooked in the vast amount of literature that continues to be published on this topic.

Authors' reply #4: We appreciate the reviewer's suggestion. However, as the detailed mechanism remains to be elucidated in the upcoming years, we believe that including such a scheme could potentially be misleading in the future. Therefore, we have refrained from adding it to the manuscript.

#5) 3. VcSduA

The discovery of a novel Shedu variant is indeed interesting. However, previous Shedu structure have shown a tetrameric state - have the authors attempted folding VcShuA as a tetramer? The authors speculate that GHKL is a sensor, but it is unclear what this is a sensor of? In other systems such as Hsp90 or DNA gyrase, GHKL functions as a clasp that opens or closes depending on bound nucleotide. I appreciate that it may be premature to speculate on the activator of VcSduA, but I would refrain from referring to this domain as a sensor.

Authors' reply #5: Indeed, a model of the putative VcSduA tetramer model has now been added as Extended Data Fig. 13e. However, as noted in the results, in contrast to the dimer, the tetramer model is not well-supported. At this point, it is difficult to speculate on what is being sensed, as excluding the shared PD-(D/E)xK nuclease domain (DUF4263), the Shedu systems studied to date have completely different N-terminal domains to the system described here. However, as detailed below in point#14, we now provide new data showing the activity of VcSduA against a vibriophage in its native 7PET *V. cholerae* that will undoubtedly aid our future studies on this topic. Finally, in response to the reviewer's concern, we have modified the text to

refrain from referring to the GHKL domain as a sensor, and have rewritten this section to better place our results in context with the findings of Gu *et al.*, 2025 (PMID: 39742666) and Loeff *et al.*, 2025 (PMID: 39742808), which showed that the N-terminal domains likely act as regulators to control the activity of the C-terminal nuclease.

#6) 4. GrwAB

Did the authors predict structures of GrwAB complexes with alternative stoichiometries? The authors conclude that GrwA and GrwB function together, and should include a supplementary figure to show these predictions, even if they are not predicted with high confidence.

Jack Bravo

Authors' reply #5: Indeed, based on our findings that when expressed individually neither GrwA nor GrwB displayed anti-phage activity, and likewise that site-directed mutants targeting either protein led to loss of function, we concluded that GrwA and GrwB likely work together. As suggested, we have now expanded this point by adding AlphaFold predicted models of the 1:1 and 2:2 GrwAB complex, both of which appear well supported (Extended Data Fig. 10e). However, we refrained from modelling alternative stoichiometries since we are not convinced that including such models would add to the current study, and elucidating the exact mechanism will require follow-up work.

Reviewer 2 (Remarks to the Author):

#7) Here, Adams *et al.* presents a very nice manuscript describing novel phage defense systems in the West African Sout American (WASA) pandemic *Vibrio cholerae* (Vc) strains. The most exciting finding of the authors is that the WASA lysogenic prophage characteristic of these strains encodes a novel phage defense systems that the authors labeled WonAB. They show that WonAB consists of a putative nuclease and ATPase and present some data that suggests this system inhibits translation. However, the mechanism of this system is still not well worked out as the authors do not perform any biochemistry to analyze the activity of these proteins nor define the targets. WonAB does protect Vc against ICP-1 phage, which is an exciting finding as this phage seems to be the major phage predator of Vc during infection. The authors also describe two other new phage defense systems encoded on the VSP-II island and identify *E. coli* phage that are sensitive to these systems. But these systems are similar to others that have already been described, and the authors do not really characterize any mechanism nor role of these systems against cholera phage. Thus, the experiments are quite well done and clearly explained, and the findings are an important contribution to the field, but my enthusiasm is somewhat dampened by a lack of understanding of WonAB function and less novelty with the two VSP-II phage defense systems. Specific comments:

Authors' reply #7: We thank the reviewer for their kind remarks about the manuscript, including their recognition of its quality, the exciting discovery of the WASA lysogenic prophage, the protection conferred by WonAB against the predatory ICP1 phage, and the well-executed experiments. We also greatly appreciate the reviewer's accurate summary of our work.

Regarding the points of critique, we fully agree that the detailed mechanisms are not described in this manuscript. However, this was not the primary aim of our study. Instead, our focus is on the biology of the human pathogen *V. cholerae*, the resistance of the WASA lineage to ICP1 phages, and the potential connection between the newly identified defence systems and the pathogen's global transmission.

Additionally, although the newly discovered VSP-II encoded systems contain domains that are found in known systems, this is not unexpected. Indeed, as recently discussed by Rousset & Sorek 2023 (PMID: 37030143) evolutionary shuffling between sensor and effector modules can diversify systems to alter their selectivity or restore activity against phages that have evolved to inhibit the system. Furthermore, and as detailed below in points #12 and #14, GrwAB and VcSduA differ significantly in both their domain composition and function compared to known systems. For instance, although GrwA resembles GmrSD, it only exhibits anti-phage activity in the presence of its novel partner protein GrwB, which shows only limited homology to GmrSD. Likewise, although VcSduA shares the common DUF4263 nuclease domain that defines the ShedU family, the N-terminal GHKL ATPase domain is completely different from those of the ShedU proteins studied to date, and to our knowledge has not been characterised in other defence systems. Moreover, as detailed below in point #14 we now demonstrate VcSduA is active in its native *V. cholerae* host against the potentially widespread vibriophage - X29.

#8) 1. Line 74-the wordings.”. broadly conserved genome encoding proteins...” is unclear. Please clarify.

Authors' reply #8: We agree that the phrasing was not intuitive and have revised it accordingly.

#9) 2. Lines 132-133-I am curious if these delayed lysis cells release viable phage. The next sentence implies that the culture rebounded due to the evolution of ICP-1 resistance, which suggests that perhaps they do and ICP-1 infection is not completely inhibited.

Authors' reply #9: We thank the reviewer for this interesting idea. As described in this paragraph, WASA-1-deficient cell begin to lyse approximately 20 minutes post-infection, whereas the non-growing WT cells persist for several hours before eventually undergoing lysis. Importantly, however, as shown in Extended Data Fig. 1d and Fig. 1e, liquid replication assays showed no evidence of ICP1 replication in WT cultures. Likewise, as shown in the new Extended Data Fig. 1c, re-streak tests conducted directly on the plaque assay plates also failed to detect ICP1 propagation. There is therefore no indication that viable ICP1 phages are released by WT cells. Instead, we think the delayed lysis phenotype likely reflects the combined effects of the initial damage sustained during ICP1 host-cell takeover (e.g. see Fig. 1d) and the inability to repair this damage due to the activation of WonAB.

Regarding the evolution of ICP1 resistance, we acknowledge that our original phrasing was unclear. In reality, the O1-antigen mutants do not evolve during the course of the experiment; rather, they spontaneously pre-exist in the population and are selected for by the

phage. This phenomenon has previously been reported by multiple researchers for *in vitro* experiments e.g. Seed *et al.* 2011 (PMID: 21304168) and Beckman & Waters 2023 (PMID: 37594274). However, such O-antigen mutants are only rarely observed *in vivo* and are actively selected against due to their susceptibility to antimicrobial peptides produced by the host immune system (Seed *et al.*, 2012 PMID: 23028317). We have rephrased these sentences for clarification.

#10) 3. Line 137- Have the authors tried to isolate escape mutants in *E. coli* using the BASEL phage?

Authors' reply #10: Despite carefully inspecting the plaque assay plates for BASEL phage able to overcome WonAB, we saw no evidence for any spontaneous escapees across multiple experiments. However, we did not conduct as extensive an effort to isolate escape mutants as we did for the *V. cholerae*-ICP1 host-phage pair, where multiple researchers attempted to isolate escapees using various methods.

#11) 4. A straightforward experiment to test if WonAB inhibits translation more directly is to add purified WonAB to an *in vitro* translation assay. If it targets tRNAs, it should block translation.

Authors' reply #11: We appreciate the reviewer's comment, which aligns with our own thoughts. Unfortunately, the process has not been straightforward. Attempts to purify the two proteins, particularly WonB, have so far been unsuccessful. Moreover, it may not be straightforward to recapitulate the phage-independent activation of WonB by WonA *in vitro*. For these reasons, we have decided to proceed with the in-depth mechanistic aspects, including *in vitro* reconstitution, as part of a follow-up work. This approach is commonly employed for newly identified defence systems, such as for example for the PARIS system (Rousset *et al.*, 2022 *Cell Host Microbe*; Burman *et al.*, 2024, *Nature*; Deep *et al.*, 2024, *Nature*) or by the DdmDE system that we recently identified (Jaskólska *et al.*, 2022 *Nature*; Bravo *et al.*, 2024, *Nature*; Loeff *et al.* 2024, *Science*; Yang *et al.*, 2024, *Cell*; Huang *et al.*, 2024, *Cell Res.*).

#12) 5. The discovery of a new Type IV restriction system in the VSP-2 island is a novel contribution. But, considering it is similar (although not identical as the authors describe) conceptually and in many ways structural (again not identical but similar), I would not consider this a high impact finding. Moreover, the authors do not provide any evidence for the role of this system in the evolution of Vc or the WASA lines, which, considering this is a major thrust of the manuscript, dampens my enthusiasm for the impact of the finding.

Authors' reply #12: In agreement with the reviewer's comments, GrwAB is similar but not identical to the GmrSD-like modification dependent restriction systems including BrxU and SspE, which consist of a single protein composed of DUF262 NTPase and DUF1524 HNH nuclease domains. However, despite the fact that GrwA shares this same domain architecture it

is unable to function in phage defence without its partner protein GrwB. Notably, GrwB has limited homology to other GmrSD-like proteins sharing only the DUF262, followed by multiple domains of as yet undetermined function. Furthermore, GrwAB is able to recognise two radically different types of DNA modification (*i.e.* 7-deazaguanine modified guanosines or sugar-modified hydroxymethyl cytosines). Thus, in our view the similar but not identical nature of GrwAB to previously studied examples of GmrSD is actually a significant strength because it raises a series of new and interesting questions such as: (i) why is GrwA unable to function alone? (ii) what is the role of GrwB? (iii) how is selectivity achieved? In our view, the discovery of GrwAB will provide new opportunities for ourselves and other researchers to better investigate how modification-dependent systems like GmrSD function and in particular how they achieve the observed selectivity for diverse DNA modifications. Finally, as we note in the results, the discovery of GrwAB together with our recent discovery of TgvAB (Vizzarro *et al.*, 2024 PMID: 39133004), hints at a potentially larger class of inactive GmrSD-like proteins that rely on partner proteins for activity.

#13) 6. Do any of the known Vc phages encode a DpdA ortholog that might be targeted by the Type IV system?

Authors' reply #13: We thank the reviewer for this comment. While we had previously examined this aspect, we have now conducted a more systematic analysis. Specifically, we inspected all GenBank and RefSeq phage sequences associated with *Vibrionaceae* as host organisms, combining the resulting 117,034 protein sequences from 1,700 assemblies into a local database. This database was searched using BLASTp for homologs of the DpdA protein from phage Bas24. The search yielded 19 hits, none of which corresponded to phages using *V. cholerae* as a host organism; instead, they were associated with other *Vibrio* species. This analysis has now been included in the revised manuscript.

Assuming our search encompassed all known *V. cholerae* phage sequences, it appears that no characterized *V. cholerae* phage harbors this modification. However, given that very few phages have been identified and characterized using *V. cholerae* as a host, this finding is not entirely surprising, in our opinion. Instead, it highlights the fact that we still lack significant information regarding *V. cholerae*-phage interactions.

#14) 7. Similar to my comment #5, the discovery of the Vc SduA system is interesting, but I would not regard it has high impact as these systems are known and there is not really any new knowledge beyond the discovery of this system in Vc and the authors do not connect the presence of this system to protection to any Vc phage.

Authors' reply #14: It is important to note that VcSduA is significantly different to the Shedu studied to date, and contains a completely different N-terminal domain (GHKL ATPase) compared to the non-enzymatic DNA-binding N-terminal domains present in the characterised systems (Gu *et al.*, 2025 PMID: 39742666; Loeff *et al.*, 2025 PMID: 39742808). Indeed, Shedu has recently been re-classified to consist of a shared C-terminal DUF4263 nuclease that can be

accompanied by highly diverse N-terminal domains predicted *in silico* to form eight broad classes of domain. Thus, our genetic discovery, characterisation and mutagenesis of VcSduA and its anti-phage activity provides the first validation of a novel class of Shedu with an enzymatic N-terminal domain, and provides a solid foundation for continued exploration of this exciting new class of Shedu, in particular the function of the GHKL domain. We recognise that this fact was not clear in the original version and have rewritten the relevant paragraph to better place our findings in the context of this new classification scheme.

Furthermore, in the revised manuscript we now extend our work on VcSduA to show that it is active at native expression levels in its native *V. cholerae* host against the vibriophage X29 (Fig. 5) and discuss the potential evolutionary advantages for the WASA lineage. Notably, phage X29 appears to be a common threat to *V. cholerae* based on the abundance of X29-targeting spacers in the CRISPR-Cas systems of 6th pandemic classical and non-pandemic strains. Interestingly, CRISPR-Cas systems are totally absent from the primary chromosomes of 7PET strains.

#15) 8. Given that WonAB is on the WASA prophage and it protects against ICP-1, why wasn't it maintained in other Vc pandemic spreading events? Does it protect against current ICP-1 phage or did ICP-1 evolve resistance or countermeasures?

Authors' reply #15:

Interestingly, rapid turnover in the mobile genetic element content of closely related strains has been observed in natural *Vibrio* populations and is likely driven by the ability of phage to overcome the defence systems carried on these elements (Hussain *et al.*, 2021 PMID: 34672730; Piel *et al.*, 2022 PMID: 35760840). Indeed, in a seminal study from the Seed laboratory, time-shift experiments using Bangladeshi patient samples showed temporal shifts in the defence system composition of SXT-ICE, which the authors demonstrated is driven by a co-evolutionary arms race between these elements and ICP phages (LeGault *et al.*, 2021 PMID: 34326207). They also found a significant fraction of *V. cholerae* remain SXT-ICE negative. Similar evolutionary sweeps have also observed with the PLE (Angermeyer *et al.*, 2022 PMID: 35164562). These reports highlight that the gain and loss of mobile genetic elements in *V. cholerae* is not uncommon and is part of the ongoing phage-host arms race. Thus, while this is an interesting question, there may be many reasons why WASA-1 prophage was not maintained, and so answering this question clearly goes beyond the scope of the current work. In our view, the simplest explanation for the dominance of SXT-ICE in current 7PET *V. cholerae* is that they are highly versatile, providing defence against ICP1, 2, and 3, as well as encoding multiple antibiotics resistances (LeGault *et al.*, 2021). Nevertheless, as described in the results, WASA-1 is found in other diverse *Vibrio* species (Supplementary Table 2). Furthermore, as described in the revised manuscript, we have now found WASA-1 (99.7% identity to Peruvian isolates from 1991) in a recently sequenced non-7PET *V. cholerae* strain isolated in Switzerland in 2019 from a patient with a travel history to Morocco, suggesting that the WonAB-containing WASA-1 continues to circulate in Africa.

Concerning the protection against more recent ICP1 phages, we have now tested eleven additional ICP1 isolates, kindly provided by Andy Camilli and Afsar Ali, including eight isolates

from the Democratic Republic of the Congo (DRC, Africa), as reported by Alam *et al.* 2022 (PMID: 36417939). Notably, WASA-1 provided protection against all of these phages, with no indication that ICP1 has evolved resistance to WonAB in these isolates. The new data have been included in Extended Data Fig. 1b.

Reviewer 3 (Remarks to the Author):

#16) There is a surge of discoveries of defense systems used to protect against bacteriophage infection, with many encoded on prophages within bacterial genomes, as reviewed recently (Murtazalieva et al 2024 <https://doi.org/10.1016/j.tim.2024.05.005>; Patel & Maxwell, 2023 (Ref 30) <https://doi.org/10.1016/j.mib.2023.102321>; Millman et al , 2022. <https://doi.org/10.1016/j.chom.2022.09.017>; Rocha & Bikard, 2022. <https://doi.org/10.1371/journal.pbio.3001514>). The manuscript by Adams et al describes the characterization of three bacteriophage defense systems in a lineage of the bacterium *Vibrio cholerae*, which can cause cholera pandemics. The authors challenged several Peruvian *V. cholerae* strains with three lytic phages ICP1-3, and discovered they could not be infected by standard plate plaque assays. The Peruvian strain behave similarly to other strains with known phage defence mechanisms, and have encode two previously described genetic loci – a VSP-II locus and a prophage called WASA-1. They discovered the WASA-1 prophage prevents ICP1 phage infection and encodes two genes WonA (a nuclease) and WonB that may function together (WonAB) as a novel defence mechanism, with homologs found in a small number of other bacteria. The authors propose WonAB promotes abortive infection: altruistic bacterial cell death that leads to fewer viruses produced. The manuscript also includes more brief description of two other loci found in the VSP-II locus: of a restriction modification system (GrwAB) and a Shedu defence system (VcSduA). Both are also found in ~1% of other bacteria. The authors are well qualified to conduct these experiments. The technical aspects of the manuscript are conducted carefully with genetic manipulation, bioinformatics, microscopy and video documentation.

Authors' reply #16: We thank the reviewers for their detailed summary of our work and for placing it in the context of the listed reviews. We also appreciate the kind comment regarding the carefully conducted experiments.

#17) The impact of the manuscript is perhaps diminished by the recent dearth of manuscripts in this area. Also, there are many important recent references lacking that can bolster the manuscript and add an ecological and evolutionary perspective as described below.

Authors' reply #17: We respectfully disagree with the comment regarding the impact of this study. To our knowledge, no other manuscripts have addressed the specific topic of this work—the biology of defence systems in *V. cholerae* and their potential link to a particular lineage of the 7PET strains. This gap became particularly evident during a recent meeting at the Royal Society in London on 'The Ecology and Evolution of Microbial Immune Systems,' where it was highlighted that studies focusing on the biology of defence systems—especially within their

natural host bacterium and tested at native expression levels (as opposed to overexpression from multi-copy plasmids)—are significantly lagging behind compared to mechanistic studies in heterologous organisms like *E. coli* or *B. subtilis*. Thus, we strongly believe that our study will contribute valuable insights to the field of bacterial defence systems, as well as to the *V. cholerae* research community, including those investigating the global transmission of this important pathogen.

Concerning the second point, namely that 'there are many important recent references lacking that can bolster the manuscript and add an ecological and evolutionary perspective,' we generally agree. However, we kindly remind the reviewers that this is not a literature review, and our manuscript must adhere to the journal's formatting guidelines. For instance, the journal's instructions to authors specify, 'References – as a guideline, we typically recommend up to 50,' and 'Main text – up to 3,500 words.'

Our manuscript already reached the envisioned length before revision and includes 107 citations (66 of which were initially cited in the main text). While we discussed with the editor some minor flexibility regarding the length of the manuscript (and potentially the number of references??), it is not feasible to expand the text further to broadly cover all related fields, such as defence systems, *V. cholerae* biology, cholera transmission, prophage biology, and ecological and evolutionary perspectives. We appreciate the reviewers' understanding that this limitation is beyond our control.

Major comments

#18) Line 127. How do the results presented at high MOI distinguish between abortive infection and killing by the phage?

Authors' reply #18:

We thank the reviewers for this important question. Indeed, as noted by LeRoux *et al.*, 2022 (PMID: 35725776):

“Abi mechanisms are traditionally thought to result from a defence mechanism that directly kills the host cell, but can also arise if the defence mechanism targets the virus, with the host cell dying because the virus triggers irreversible damage, such as chromosome degradation. One key characteristic of Abi mechanisms is that when most cells are infected at a high multiplicity of infection (MOI), the growth of the bacterial population stops, while at lower MOIs, the uninfected bacteria can continue to grow.”

Thus, while the data we present in Fig. 2 and Fig. 3 strongly support the interpretation that WonAB is an abortive infection system that aborts ICP1 infection by inhibiting translation and thus preventing the progression to structural protein assembly, our data do not distinguish between cell death as a result of WonAB activation compared to the actions of ICP1. Notably, as shown in Fig. 2d WonAB does not prevent the initial the damage (Fig. 2d) associated with the rapid takeover of by ICP1 (McKitterick *et al.*, 2019 PMID: 31600502; Barth *et al.*, 2020

PMID: 33051375). We have therefore updated the text to reflect the fact that the cell is likely irreversibly damaged during the initial stages of ICP1 infection.

#19) Fig 1A and 1C. The plaques do not dilute out serially on the Peru strains. The authors might consider assessing the phage population over time in liquid cultures as described in <https://doi.org/10.1101/2024.07.10.602838>.

Authors' reply #19: The zones of partial lysis shown on these panels are unlikely to represent true phage plaques. Indeed, liquid replication assays (Fig. 1e and Extended Data Fig. 1d) assembled using a similar setup to that in Alseth *et al.*, 2024 (10.1101/2024.07.10.602838) already showed no evidence of ICP1 propagation despite infection proceeding similarly to the ICP1-susceptible controls, which exhibited a >4-log increase in ICP1 within 2 hours. Nevertheless, to address this question more directly we have now performed re-streak tests directly on the plaque assay plates, by taking material from drop exposed to the highest concentration of phage and re-streaking on an ICP1-sensitive indicator strain. Notably, as discussed in the revised result section and shown in the new Extended Data Fig. 1c, these re-streak tests also failed to detect ICP1 propagation. There is therefore no indication that viable ICP1 phages are released by WT cells. Instead, these zones of partial lysis likely represent “lysis from without” *i.e.* lysis at high phage concentration without viable phage production, as previously observed by Hussain *et al.*, 2021 (PMID: 34672730).

#20) Fig 2A. What was the rationale for running the time-course for only 5 hours (300 minutes)? Might you expect recovery if this was measured for longer (ex: 24h)?

Authors' reply #20: Plate reader experiments were run for 5 hours because this time period captures the collapse of the unprotected Δ WASA-1 cultures, the protection of the population observed at low MOI and the growth arrest at high MOI in the ICP1-resistant WT cultures. In addition, this time period is already sufficient to capture the recovery of the cultures due to the selection of pre-existing O1-antigen mutants in the population, which as detailed above under reply #9 has previously been observed by multiple researchers. Thus, in our view further extending the time-frame of this experiment would be unlikely to generate any meaningful new information.

#21) Fig 3. Why was the experiments terminated at 3 hours when the hypothesis is that WonAB inhibits translation and does not induce lysis. Was the experiment extended for longer time periods?

Authors' reply #21: This experiment was terminated at 3 hours, as we did not anticipate gaining additional information by extending its duration. However, we have now repeated the experiment with an extended 6-hour timeframe, but still did not observe significant lysis nor the did the bacteria revert to normal growth. Importantly, even if recovery were to occur under these

conditions, which only mimic the translational inhibition triggered by WonAB activation, it would not happen during actual phage infection, as ICP1 takes over the cell within as little as 4 minutes (see review by Boyd *et al.*, 2021 PMID: 34314595).

Minor comments:

#22) Line 18 replace “predatory” with “lytic”

Authors’ reply #22: The term 'predator' is commonly used in the ICP1 literature e.g. Boyd *et al.*, 2021 'Bacteriophage ICP1: A Persistent Predator of *Vibrio cholerae*' (PMID: 34314595). Therefore, we prefer to retain the term 'predatory' in our manuscript.

#23) Line 26 please provide citations

Authors’ reply #23: This part has been updated to: ‘Bacteriophages also have the potential to affect bacterial pathogenesis, as exemplified by the cholera toxin-encoding prophage CTX Φ^2 . Moreover, bacteriophage predation is thought to limit the duration and severity of cholera epidemics, as well as affecting individual patient outcomes³⁻⁶’.

#24) Line replace “predation” with “lysis”

Authors’ reply #24: See comment #22 above (please note that no line number was indicated).

#25) Line 32 – Why is there an urgent need? Is evolved resistance to phage observed? Common? If yes, what kind of resistance? Surface receptor-based?

Authors’ reply #25: If as proposed, phages are to be used to prevent cholera transmission towards household contacts (Yen *et al.*, 2017 PMID: 28146150), there is an urgent need to better understand how *V. cholerae* can resist phage predation in general. We believe this is well reflected in this and the preceding sentence. Notably, this isn’t specific to any particular mode of resistance (e.g. receptor-based, as mentioned by the reviewers) but is instead comparable to antibiotic resistance when antibiotics are used for treatment. Furthermore, an understanding of the anti-phage defence systems active against these phages, and the mechanisms that the phages use to overcome these systems will be crucial to informing future efforts to select (and potentially engineer) phages suitable for phage therapy.

#26) Line 42 – “...with the emergence ...” is confusing and should be reworded. In particular “resistant ICP1” is awkward since phage resistance is commonly ascribed to the bacterial host not the virus. We suggest rewording to clarify what is being described is the emergence of ICP1 capable of infecting a phage resistant host.

Authors' reply #26: Agreed. This sentence has been rewritten as follows: '...with the emergence of resistant ICP1 phages that can overcome these defence mechanisms selecting for either alternative SXT-ICE carrying new defence systems or new PLE variants^{11,14}.'

#27) Line 45 –A review of host-phage interactions (e.g. <https://doi.org/10.1038/s41564-024-01832-5>) would be useful for a reader here in the introduction. Prior work suggests defenses play a minor role compared to surface modifications. Have *Vibrio* surface modifications been reported that promote phage resistance?

Authors' reply #27: The topic of what drives phage-host range remains a highly active area of research. In the suggested article by Gaborieau *et al.*, 2024 (PMID: 39482383) the authors examine natural isolates from the genus *Escherichia* and conclude that most interactions are explained by adsorption factors as opposed to anti-phage defence systems. However, previous work in other bacteria, for example Hussain *et al.*, 2021 (PMID: 34672730); Piel *et al.*, 2022 (PMID: 35760840) and Costa *et al.*, 2024 (PMID: 38394193), reach differing conclusions on the role of defence systems, which likely reflects species-specific differences and the analysis of co-evolving bacteria-phage pairs.

Furthermore, prior work from the teams of Kimberley Seed, Andrew Camilli, has demonstrated that the ICP1 receptor (the O1 antigen) mutates frequently *in vitro* with these mutants being subsequently selected for by ICP1 predation. But this scenario does not occur *in vivo* due to the susceptibility of these receptor-mutant strains to the innate immune response (and potentially also other environmental insults). Thus, the receptor versus defence systems discussion is highly context-dependent and not the primary focus of our study.

For these reasons, we will not delve further into this topic beyond the already included information regarding the O1 antigen as a receptor and its mutability *in vitro*. We hope the reviewers agree with our decision to maintain the focus of our manuscript.

#28) Line 59 – Did the authors look for any evidence of phenotypic changes that might result from modification of cell surface receptors?

Authors' reply #28: We are somewhat confused by this comment. The majority of the manuscript focuses on the underlying reason why the strains are resistant to ICP1, conclusively demonstrating that this is due to the WonAB system and not the O1 antigen as a surface receptor. Importantly, we also show that the O1 antigen remains intact in these strains, as evidenced by the fact that ICP3 plaques on the WASA lineage strains in a manner similar to that observed for non-WASA 7PET strains (Fig. 1a). Therefore, we strongly believe this comment has been more than sufficiently addressed in the manuscript.

#29) Line 137, 223. Perhaps for an audience less familiar with ICP1, the author could remind readers that ICP1 is a phage. For example, consider rewording to replace "ICP1 escape

mutant”, “escapers” and “escaper mutants” with “ICP1 escape phage”. This would remind the audience that these mutations are in the phage not the bacterial genome.

Authors’ reply #29: As suggested we have modified the text to emphasise that the escape mutants of the named phages are mutants able to overcome the defence system.

#30) Line 105 and 118-120. WonAB is found in 0.43% of bacterial genomes. Similarly, line 233 describes GrwAB as “widespread”? At 1%? In line 260 VcSduA homologs are noted as “widespread”. At ~1.2%? A frank recognition that the systems described are relatively rare is warranted.

Authors’ reply #30: The term widespread was meant to reflect the widespread distribution of the systems in different bacterial orders, not their abundance (Extended Data Figs. 3f, 10f, 13f). To avoid any confusion, we have removed the terms ‘widespread’ and widely distributed’ from the text, as suggested. Notably, since classification of defence systems families remains an evolving topic, we have refrained from making statements regarding the abundance of these systems and instead provide the numbers and complete lists of homologues identified to allow readers to judge the findings for themselves.

#31) Lines 133-136. The authors should unpack this result more. Is the candidate defence system simply delaying lysis and allowing for emergence of spontaneous O1 antigen receptor mutants? What do the authors mean by *in vitro* here? Outside the human host? In the laboratory? Are O1-antigen mutants observed in *V. cholerae* derived from non-pandemic sources?

Authors’ reply #31: As also described above in reply #9, we acknowledge that our original phrasing regarding the evolution of ICP1 resistance was unclear. In reality, the O1-antigen mutants do not evolve during the course of the experiment; rather, they spontaneously pre-exist in the population and are selected for by the phage. This phenomenon has previously been reported by multiple researchers for *in vitro* experiments e.g. Seed *et al.* 2011 (PMID: 21304168) and Beckman & Waters 2023 (PMID: 37594274). However, such O-antigen mutants are only rarely observed *in vivo* and are actively selected against due to their susceptibility to antimicrobial peptides produced by the host immune system (Seed *et al.*, 2012 PMID: 23028317). We have rephrased these sentences for clarification. Furthermore, we use the terms '*in vitro*' (outside the host) and '*in vivo*' (inside the host) as standard nomenclature in the field, and we do not believe these require additional definition for the readership of this journal.

Concerning the question 'Are O1-antigen mutants observed in *V. cholerae* derived from non-pandemic sources?': O1 *V. cholerae* are generally very rare in non-pandemic contexts, and we are not aware of any O1 antigen mutants that were isolated from this setting.

#32) Line 192 This section beginning here would benefit from a brief reminder of the initial hypothesis to explore VSP-II.

Authors' reply #32: We strongly believe that the first sentence of this paragraph, which begins with 'The second genetic signature of the WASA lineage is a unique variant of VSP-II (Fig. 4a),...' sufficiently reminds the reader of the rationale for exploring VSP-II.

#33) Line 243. This section on VcSduA would benefit from an introduction.

Authors' reply #33: As mentioned above, the manuscript had already reached its length limit before incorporating the new revision experiments and information. Therefore, we prefer to keep this section concise, as it does not affect the understanding of the subsequent description of the results. To aid the reader we have now added additional subheadings to introduce the different sections.

#34) Lines 265-266. Some context would be helpful here. This section reads more like a list - we found two more.

Authors' reply #34: As outlined in further detail above in reply #14, we recognise that the relationship of VcSduA to the Shedu family was not clearly explained in the original version and have now rewritten this paragraph to better convey our findings.

#35) Line 276. It is fascinating that many anti-phage systems are encoded on prophages. The impact of the manuscript could be improved by testing (comments for lines 141-142) and discussing the implications of WonAB being on the WASA-1 prophage itself. The lifecycle of the well studied *V. cholerae* CTX phage has profound effects on the dynamics of the *Vibrio* population as well as the phage itself. A more in-depth exploration of the consequences of this arrangement would be a welcome addition.

Authors' reply #35: We agree with the reviewers that it is fascinating how many phage defence systems are encoded on prophages. However, delving into prophage biology is clearly beyond the scope of the current manuscript but will be addressed in follow-up work.

#36) Line 287. The authors should draw on a wealth of citation to discuss how prophages and the genetic material they carry can help their bacterial hosts "succeed".

Authors' reply #36: As mentioned above, the journal's length limitations do not allow us to expand further on prophage biology beyond what is already included, where we include three citations covering the role of prophages in the context of anti-phage defence systems.

#37) Line 281. Citations describing “superinfection exclusion” are needed here as well as use of this term in the manuscript. It is well described in the literature (ex: <https://doi.org/10.1371/journal.pcbi.1010125>) but not referred to anywhere in the manuscript.

Authors’ reply #37: We thank the reviewers for the interesting question. However, by our understanding, superinfection exclusion refers to the ability of a resident prophage to block secondary infections by the same or closely related phages. Moreover, this is typically achieved by either altering cell-surface receptors to block phage adsorption or by interfering with genome injection and entry into the cytoplasm (e.g. Patel & Maxwell 2023 PMID: 37121062; Getz & Maxwell 2024 PMID: 38950439). Given that ICP1 and WASA-1 are not closely related phages, and that WonAB is an abortive infection system that neither blocks phage adsorption nor genome entry, we therefore do not think that a discussion of superinfection exclusion is warranted.

#38) Lines 297-300. Again, the Discussion can be improved by mention of ecological and evolutionary theories regarding mobile genetic elements. Several recent reviews include: <https://doi.org/10.1016/j.mib.2024.102436>; <https://doi.org/10.1038/s41467-024-46489-0>;

Authors’ reply #38: We thank the reviewers for the helpful suggestions. However, as mentioned above, the journal's strict length limitations prevent us from expanding the discussion in the many diverse directions suggested by the reviewers without shortening the results section.

#39) Lines 141-142. Is WonAB activated in response to other stressor beyond phage? It would be interesting to know whether ICP1 induces the WASA-1 WonAB-containing prophage.

Authors’ reply #39: We thank the reviewers and completely agree that these are interesting questions. However, since the current study focuses on phage defence, these intriguing follow-up questions will need to be addressed in future work, as they fall outside the scope of this study.

#40) Lines 87-88 – Do prophages carry antiphage defence system to increase their host’s fitness? The results suggest “superinfection exclusion” (<https://doi.org/10.1371/journal.pcbi.1010125>), which selfishly benefits the prophage by preventing infection by other phage.

Authors’ reply #40: Please see author reply #37 above.

#41) The manuscript's impact can be improved by recognition and integration of current work exploring the evolutionary consequences of phage co-infection on community composition. Indeed there are two recent papers on phage dynamics in *Vibrio* populations that need to be included here: Hussain, et al 2021 (<https://doi.org/10.1126/science.abb1083>), Piel et al 2022 (<https://doi.org/10.1038/s41564-022-01157-1>).

Brian K Hammer, PhD

Ellinor O Alseth, PhD

Authors' reply #41: We thank the reviewers for the helpful suggestions. We have now added citations to the two suggested studies into an expanded discussion, placing them in the context of the evolution of new lineages of pandemic *V. cholerae* and the roles of mobile genetic elements and phage-bacteria co-evolution in this process. Finally, we hope that the reviewers can understand that given the restrictions on the manuscript length and the number of references suggested throughout this rebuttal letter, we needed to prioritise what to include as discussion points during the revision.

Response to editor & referees

We thank the editor for the positive outcome of the peer review process.

As discussed with Dr. Frischkorn, there were no remaining reviewer requests that needed to be addressed within the length-restricted text.